# Application of Biomarkers in Spinal Muscular Atrophy

**DOI:** 10.3390/ijms26146887

**Published:** 2025-07-17

**Authors:** Changyi Gao, Yanqiang Zhan, Hong Chen, Chunchu Deng

**Affiliations:** Department of Rehabilitation Medicine, Tongji Hospital, Tongji Medical College, Huazhong University of Science and Technology, Wuhan 430030, China; m202476862@hust.edu.cn (C.G.); zhanyq77@163.com (Y.Z.)

**Keywords:** spinal muscular atrophy, biomarker, clinical application, advanced technology, therapeutic strategy

## Abstract

Spinal muscular atrophy (SMA) is a fatal motor neuron disease characterized by five clinical subtypes, each presenting with different rates of disease progression and varying responses to recently approved therapies. The identification of reliable biomarkers is essential for improving diagnosis and prognosis, monitoring disease progression, enabling personalized treatment strategies, and evaluating therapeutic responses. In this review, we conducted a comprehensive literature search using PubMed and Web of Science with the keywords “spinal muscular atrophy”, “biomarker” and advanced technologies such as “single-cell omics”, “nanopore and long-read sequencing” and “epigenetics” to identify and summarize current advances in SMA biomarker discovery and application. We begin with a brief overview of SMA and its current treatment barriers. We then conclude with well-established and emerging molecular and non-molecular biomarkers, followed by a conclusion of emerging technologies in biomarker discovery. In the meantime, we highlight the application of biomarkers in key areas, including early diagnosis and disease stratification, monitoring of disease progression, and prediction of treatment response. Finally, we summarize biomarker-targeted therapies, addressing current challenges in biomarker research, with the goal of improving clinical outcomes for patients with SMA.

## 1. Introduction to Spinal Muscular Atrophy

Spinal muscular atrophy (SMA) is an autosomal recessive neuromuscular disorder marked by the progressive degeneration of motor neurons in the anterior horn of the spinal cord [1]. The observed degeneration primarily results from the protein deficiency of the survival motor neuron (SMN), caused by the absence or mutation of the *SMN1* gene, and causing progressive muscle weakness and atrophy [1]. The prevalence of SMA is estimated to be between 1/6000 and 1/10,000 in childbirth. The carrier frequency within the general population is roughly 1 in 40 to 1 in 50 individuals, indicating a significant asymptomatic carrier presence that impacts inheritance patterns and genetic counseling considerations [2,3]. This relatively uniform carrier rate across populations corresponds to a 25% risk of having an affected child for carrier couples [4]. SMA imposes a substantial disease burden, including extremely high therapeutic costs, ongoing rehabilitation needs, and significant impacts on family income and mental health [5,6]. In addition, the long-term need for specialized care, adaptive education, and accessible infrastructure places considerable demands on public health and social support systems [5,6].

The causative gene underlying SMA is in the 5q13.2 region of human chromosome 5 [7]. It is part of the highly homologous survival of the *SMN* gene family, primarily comprising the *SMN1* and *SMN2* genes [8,9]. The *SMN1* gene encodes the full-length SMN protein, a critical factor for the viability of motoneurons in the spinal cord’s ventral horn [8,9]. In contrast, the *SMN2* gene, which shares approximately 99% sequence homology with *SMN1*, harbors a single nucleotide polymorphism in exon 7 (c.859G > C) that leads to the exclusion of this exon in roughly 90% of its transcripts [10]. This results in the production of an unstable truncated protein variant known as Δ7-SMN [10]. Only 10% of *SMN2* transcripts can generate the full-length functional SMN protein [10]. The predominant genetic alteration associated with SMA, occurring in around 95% of cases, is a homozygous deletion or conversion of the *SMN1* gene to *SMN2* across both alleles on chromosome 5, resulting in a total absence of functional SMN protein [11,12]. In approximately 5% of affected individuals, point mutations or small insertions/deletions within the *SMN1* gene lead to diminished expression or production of non-functional SMN protein [13,14].

Gene therapy is one of the current treatment strategies for SMA [15]. However, therapies like Zolgensma are costly, limiting accessibility for many families. Their effectiveness depends heavily on early administration, as patients with advanced neuromuscular damage may not fully regain function [15]. While short-term safety data are available, long-term risks remain uncertain. A 2025 case reported astrocytoma development in a child post-Zolgensma treatment, though vector integration analysis suggested no causal link [16]. Antisense oligonucleotides (ASOs), such as nusinersen (Spinraza), require repeated intrathecal injections—six in the first year, then three annually—posing physical, logistical, and financial burdens [17]. ASOs can alleviate symptoms and improve quality of life, but do not reverse advanced muscle atrophy or cure the disease. Adverse effects, including cephalalgia and localized reactions at the injection site, though generally mild, may impact compliance [18]. Small-molecule drugs like risdiplam (Evrysdi) and branaplam increase SMN protein levels but are less effective in late-stage disease and cannot repair severely damaged motor neurons. Long-term safety is still under study, with possible systemic toxicities requiring clinical monitoring. Efficacy also varies across individuals due to genetic and disease-related factors.

Early diagnosis, prognosis assessment, treatment monitoring, and the use of surrogate endpoints in clinical trials are critical to advancing therapeutic strategies for SMA. As technological innovations progress and global attention to rare diseases increases, SMA has gained greater recognition, leading to the identification of potential biomarkers. This review provides a brief overview of SMA and discusses validated biomarkers, those under clinical investigation, and emerging technologies driving biomarker discovery. We then explore the clinical applications of biomarkers in early diagnosis, disease stratification, progression monitoring, treatment response prediction, therapeutic guidance, and biomarker-targeted interventions. In conclusion, we examine the contemporary challenges and potential avenues for future research in this domain, focusing on biomarker research to improve clinical outcomes for individuals with SMA.

## 2. Molecular Biomarkers for Spinal Muscular Atrophy

Based on previously published data, we identified both established and emerging biomarkers for spinal muscular atrophy, along with their potential clinical applications (Table 1).

### 2.1. Survival Motor Neuron

SMA is caused by the absence or mutation of the *SMN1* gene. The deletion or mutation in the *SMN1* gene is the primary diagnostic biomarker, combined with clinical signs [1,10,19]. Humans typically carry one to four copies of the *SMN2* gene, with two copies being most common [10]. Due to a critical C-to-T transition in exon 7, *SMN2* predominantly produces a truncated, non-functional SMN protein, with only a small proportion of full-length protein being generated [10]. The number of *SMN2* copies serves as the primary genetic modifier of SMA severity: higher copy numbers are associated with increased production of functional SMN protein and correspondingly milder disease phenotypes [10].

Clinically, SMA is categorized into five types (0–4), with disease severity closely linked to *SMN2* copy number [13]. The most severe form, type 0, typically corresponds to a single *SMN2* copy [118], while type 1 patients usually possess one or two copies, type 2 three copies, type 3 three to four copies, and type 4 four or more copies [10]. Complete absence of both *SMN1* and *SMN2* leads to embryonic lethality, highlighting the essential compensatory role of *SMN2* in SMN protein synthesis and survival [118].

Although the majority of SMA patients have between one and four *SMN2* copies, rare cases with up to eight copies have been documented [13,118]. There is a well-established inverse correlation between the number of *SMN2* gene copies and the severity of clinical presentations in affected individuals (Table 2) [13,119,120]. Notably, asymptomatic individuals with homozygous *SMN1* deletions but multiple *SMN2* copies have been identified, further supporting the gene’s protective role. This genotype-phenotype relationship has been validated in transgenic Smn–/– mouse models carrying varying copy numbers of human *SMN2* phenotype [23,24,25]. Mice with only one or two *SMN2* copies exhibit rapid motor neuron loss and early mortality, mimicking human SMA type I [26,27]. In contrast, mice harboring eight *SMN2* transgene copies show preserved motor neurons, extended survival, and improved motor function [26,27]. These findings underscore the dose-dependent neuroprotective effect of *SMN2* and its central role in modulating SMA disease progression [13,26,27,119,120,121].

Multiple studies investigating pharmacologic treatments for SMA have shown that patients with higher *SMN2* copy numbers tend to respond better to therapy. Therefore, the copy number of *SMN2* is suggested to predict the treatment response in SMA [20]. Importantly, this relationship and the supporting evidence primarily derive from studies in infants and children, where the impact of disease duration is less pronounced. For example, the degree of improvement in CHOP INTEND and HINE-2 scores from baseline to 12 months after initiating the treatment of nusinersen is highly influenced by *SMN2* copy number [28]. CHOP INTEND and HINE-2 scores were used to quantify motor performance and track treatment response in infants with SMA [28]. In a cohort of children with SMA treated with nusinersen before age 3, those with 2 *SMN2* copies exhibited poorer motor, respiratory, and orthopedic outcomes after 36 months of treatment compared to those with 3 copies [29]. Although SMN levels are commonly used in research and clinical settings, they do not serve as a consistent biomarker for tracking disease progression or assessing treatment efficacy in SMA, and their utility may differ between infants/children and adults with longer disease duration. Xing et al. noted several limitations [30]. SMN transcript and protein levels in peripheral blood show significant variability and overlap between SMA patients and healthy controls and are generally lower than those in motor neurons [31,32,33,34]. SMN levels in CSF are not reliably quantifiable, and monitoring central nervous system (CNS) protein levels in living subjects [33,35]. For instance, mRNA levels of SMN in blood maintained a state of relative constancy in a postnatal pig model exhibiting reduced expression levels of SMN in motor neurons [36]. Other studies have shown that some non-specific therapies can also affect SMN levels, suggesting that SMN expression is influenced by multiple factors [37]. Restoration of SMN levels does not necessarily correlate with improvements in motor function [38,39]. For example, a study conducted by Chiriboga et al. indicated that intrathecal nusinersen is well-tolerated in pediatric patients aged 2 to 14 with SMA types 2 and 3, with no significant safety issues reported. CSF concentrations of the drug exhibited a proportional increase relative to dose (ranging from 1 mg to 9 mg) [38]. At the 9 mg dosage, a statistically significant and sustained enhancement in motor function, as assessed by HFMSE scores, was noted in this pediatric cohort, with a 3.1-point increase observed at 3 months post-administration (*p* = 0.016). This improvement further escalated to 5.8 points at the 9–14 month follow-up (*p* = 0.008) [38]. In terms of SMN protein levels in CSF, a more than twofold increase was observed in both the 6 mg and 9 mg cohorts over the 9–14 month period, but it was not statistically significant [38]. Yeo et al. [40] pointed out that although *SMN2* copy number remains the only genetic predictor of SMA severity, genotyping results can vary by up to 45% between different laboratories, and even within the same laboratory. This underscores the need for further research into the sequencing, structure, and functional role of *SMN2* in SMA, to enable more precise use of costly and often lifelong therapies across all age groups, particularly considering the potential differences in adults with long-standing disease [40]. Lapp et al. also noted that although a deficiency of SMN protein and transcripts is a pathological hallmark of SMA, their evaluation in body fluids appears to have limited utility due to weak correlations with clinical features, a limitation that may be more significant in adults with longer disease duration where compensatory mechanisms and chronic pathological changes could alter the relationship between SMN levels and functional outcomes [41].

### 2.2. Neurofilaments

NFs are neuron-specific cytoskeletal proteins classified as type 4 intermediate filaments. Upon neuronal injury, NFs are released into the extracellular space, with elevated levels reported across multiple neurodegenerative disorders [42]. They exist as heteropolymers comprising three subunits of distinct molecular weights: the NfL at 68 kDa, NfM at 150 kDa, and NfH at 200 kDa [42]. In SMA mouse models, neurofilament transcripts such as NfL in the spinal cord are reduced during early symptomatic stages and partially increase at later stages. In the blood, levels of NfHSMI 34 and NfHSMI 35 proteins are elevated during both early and late symptomatic stages in SMA type 1 mice and also increased during early symptomatic stages in SMA type 3 mice. After ASO treatment in SMA-I mice, NfL-related biomarkers show short-term changes [43].

In patients with SMA type 1, especially those under four years old, elevated levels of phosphorylated neurofilament heavy chain (pNfH) have been detected in the CSF [44,45,46]. These studies found that higher baseline pNfH levels are associated with earlier disease onset and poorer motor function. Over time, pNfH levels tend to decrease toward levels seen in healthy controls. For example, a study assessed plasma pNfH as a biomarker in SMA using the ProteinSimple^®^ platform, comparing levels in infantile-onset SMA patients from the ENDEAR trial (NCT02193074) with those in non-neurological controls [47]. Baseline pNfH concentrations were markedly elevated in SMA infants (median 15,400 pg/mL), approximately tenfold higher than age-matched controls (*p* < 0.0001) and ~90-fold higher than in older children without SMA (*p* < 0.0001) [47]. Higher pretreatment pNfH levels correlated with earlier symptom onset, earlier diagnosis and treatment initiation, lower CHOP-INTEND scores, and reduced CMAP amplitudes [47]. Nusinersen treatment led to a rapid reduction in pNfH (71.9% by Month 2 and 90.1% by Month 10), significantly greater than the decline observed in sham controls (16.2% and 60.3%, respectively) [47]. These findings support plasma pNfH as a sensitive biomarker of disease burden and a dynamic indicator of therapeutic response in infantile-onset SMA, particularly suited to monitoring early treatment effects [47]. Analysis from the NURTURE study demonstrated significantly elevated levels of pNfH in both plasma and CSF of presymptomatic infants with SMA compared to healthy controls (geometric mean plasma pNfH: 1090.8 pg/mL in controls; substantially higher in SMA), with markedly higher concentrations observed in infants with two versus three *SMN2* copies (plasma: *p* = 0.0050; CSF: *p* = 0.0020) [49]. Nusinersen treatment induced a rapid decline in pNfH during the loading phase, followed by stabilization in both compartments [49]. Baseline pNfH levels were strongly predictive of motor outcomes, exhibiting significant inverse correlations with HINE-2 scores at Day 302 (rs = −0.53, *p* = 0.0120) and age at walking onset (rs = 0.55, *p* = 0.0147) [49]. Notably, pNfH levels at Day 64 emerged as the most robust predictor of subsequent motor function, correlating with Day 302 HINE-2 scores (rs = −0.67, *p* = 0.0005) and walking milestone attainment (rs = 0.64, *p* = 0.0025) [49]. These findings further establish pNfH as a sensitive biomarker of early disease activity and treatment response in presymptomatic SMA, highlighting the critical window for therapeutic intervention and reinforcing the importance of newborn screening [49]. The rapid, treatment-driven decline in pNfH suggests it reflects reversible pathophysiology, although whether this represents preclinical neurodegeneration or modifiable disease processes warrants further investigation [49].

A separate study evaluated serum NfL as a biomarker in 18 pediatric patients with SMA undergoing nusinersen treatment, and established age-specific reference values using Single Molecule Array (SIMOA) technology from 97 healthy controls (Nitz et al. [44]). Treatment-naïve SMA patients with two *SMN2* copies exhibited significantly higher median serum NfL levels compared to both patients with more than two copies (*p* < 0.001) and age-matched controls (*p* = 0.010), with levels declining over the course of treatment [44]. In healthy children, the median serum NfL concentration was 4.73 pg/mL, varying significantly with age (*p* < 0.001) but not with sex (*p* = 0.486) [44]. Among SMA patients, strong correlations were observed between serum and CSF NfL levels (r = 0.7, *p* < 0.001) [44]. Notably, serum NfL levels inversely correlated with motor function in children with two *SMN2* copies (r = −0.6, *p* = 0.134), but not in those with more than two copies (r = −0.1, *p* = 0.744) [44]. These findings support serum NfL as a promising biomarker of disease activity in early-stage SMA, particularly in patients with two *SMN2* copies, and provide valuable pediatric reference values for future clinical and translational studies.

Although pNfH and NfL show great promise as biomarkers in infants with SMA, data in adolescents and adults with type 2–4 SMA remain somewhat inconsistent. Wurster et al. demonstrated the limited utility of neurofilaments as biomarkers in chronic SMA types 2/3 [50,122]. Their studies of 25–46 adolescent/adult patients revealed CSF and serum NfL levels remained within normal ranges (mean serum NfL: 7.2 vs. 6.8 pg/mL in controls) and showed no treatment-related changes, despite inverse correlations with motor function (HFMSE ρ = −0.525 at month 10) [50,122]. Similarly, Totzeck et al. [59] found no significant alterations in four axonal degeneration markers (including NfH and tau) in 11 SMA type 3 adults during nusinersen loading [50,122]. Freigang et al.’s multicenter study (2022) of 79 SMA patients (58 adults, 21 children) identified CSF NfL as more clinically informative than serum markers [51]. CSF NfL showed strong correlations with disease severity and significant decreases during 14 months of nusinersen treatment (unlike stable serum levels), suggesting better sensitivity for monitoring treatment response across ages and subtypes [51]. In SMA mice, quantitative polymerase chain reaction (PCR) and enzyme-linked immunosorbent assay (ELISA) revealed significant alterations in NF transcript and protein levels (both NfH and NfL) in spinal cord and blood across disease stages and phenotypes (Spicer et al. [43]). While blood NfH levels effectively indicated disease onset and transiently responded to ASOs treatment in mice, despite persistent therapeutic benefits, NfH levels showed a dramatic 90% decrease during early postnatal development in both control and SMA mice (Spicer et al. [43]). In human patients, blood NfH levels were reduced in older children with chronic SMA progression. These findings demonstrate that blood NfH serves as a valuable biomarker for acute disease monitoring and treatment response in early-stage SMA, but its utility appears limited in chronic SMA cases, underscoring the need to identify additional biomarkers for long-term disease management (Spicer et al. [43]).

Furthermore, another study found no consistent treatment-related changes in pNfH or NfL levels in either CSF (measured by ELISA) or plasma (NfL by SIMOA) across 22 months of nusinersen treatment in 31 adult SMA type 2/3 patients, despite observing clinical improvements in ambulatory patients (Revised Upper Limb Module [RULM] *p* = 0.04, Hammersmith Functional Motor Scale-Expanded [HFMSE] *p* = 0.05 at 24 months) [65]. Both neurodegeneration markers remained stable throughout all measured timepoints (2, 6, 10, and 22 months), showing no significant correlations with clinical outcomes [65]. This contrasts with pediatric SMA where these biomarkers are more dynamic, likely reflecting the slower disease progression and smaller pools of actively degenerating neurons in adult forms [65]. The stability of pNfH and NfL suggests limited utility as treatment response biomarkers in chronic SMA, though their baseline levels may still provide diagnostic information about axonal status [65].

These findings suggest that conventional neurodegeneration biomarkers may reflect longstanding motoneuron loss rather than active disease in slowly progressive SMA, underscoring the need for stage-specific and compartment-targeted biomarker strategies in SMA. In summary, while NFs appear to be reliable biomarkers for predicting prognosis and treatment response to nusinersen in infants with SMA, their value in later-onset SMA and during the disease progression of all SMA types remains unclear. Further research is needed to assess the suitability of NFs as biomarkers in late-onset SMA.

### 2.3. Combinatory microRNAs

MiRNAs are a class of small regulatory non-coding RNAs approximately 22 nucleotides in length, capable of modulating the expression of complementary messenger RNAs. Among non-coding RNAs, miRNAs are the most extensively studied with regard to their role in post-transcriptional gene regulation [123]. Multiple studies have provided evidence implicating miRNAs in the pathogenesis of SMA [89,90,124]. Chen et al. reported that circulating miRNAs in peripheral blood—such as miR-1, miR-9, miR-132, miR-133a/b, and miR-206—have potential utility in disease monitoring, and further suggested that miRNAs may not only serve as biomarkers but also represent potential therapeutic targets [90]. However, this study did not specifically differentiate recommendations between pediatric and adult patients.

Studies in pediatric SMA patients, particularly infants, have identified specific miRNA dynamics. Xing et al. summarized findings from several studies indicating that, in infants with SMA, serum levels of miR-133a/b, miR-206, and miR-1 decreased after six months of nusinersen treatment [30,91]. In contrast, levels of hsa-miR-181a-5p, miR-324-5p, and miR-451a were significantly upregulated [92]. Furthermore, at two months of treatment, the expression levels of miRNAs, such as miR-107, miR-142-5p, miR-335-5p, miR-423-3p, miR-660-5p, and miR-378-3p, were predictive of treatment response at six months, with miR-142-5p and miR-378a-3p showing particularly strong predictive potential. Notably, the relative expression levels of miR-340-5p at two months were significantly associated with subsequent motor function improvement [93].

Barbo et al. highlighted that several studies support the use of serum muscle-specific miRNAs (myomiRs)—such as miR-133a, miR-133b, and miR-1—as biomarkers for monitoring the therapeutic response to nusinersen, with their expression changes correlating with clinical improvement [91,94,100]. These studies primarily involve pediatric cohorts [91,94,100]. In late-onset SMA patients, lower baseline CSF levels of miR-133a-3p and miR-206 were associated with better clinical outcomes after six months of nusinersen treatment, with miR-206 levels inversely correlating with HFMSE scores (Magen et al. [96]), while other studies found no significant changes in miR-206 levels post-treatment in pediatric SMA patients [91] or differences between SMA patients and healthy controls [99].

Additionally, certain miRNAs in CSF, including miR-132, miR-218, and miR-9, were found to be upregulated following nusinersen treatment and were associated with enhanced motor function in SMA type 1 and 2 patients [91,92,95,96,97,98,99,100,101].

Collectively, these findings suggest that individual or combinatorial miRNAs hold significant promise as biomarkers for monitoring disease progression, forecasting therapeutic response, and evaluating the efficacy of treatments in SMA, with the strongest evidence currently derived from studies in pediatric populations, especially infants. Data supporting specific recommendations for adult SMA patients based on miRNA biomarkers are comparatively sparse.

### 2.4. Cytokines

Cytokines have drawn increasing attention in studies of neuroinflammation associated with SMA. Molecules such as IL-8, IL-23, and MCP-1 have demonstrated potential as biomarkers [73]. However, cytokine profiles vary across patient cohorts and biological fluids, and these molecules interact with each other, are influenced by glial cell activity, and are temporally dependent on treatment. As such, the interpretation of individual inflammatory mediators must be approached with caution. Although no definitive cytokine biomarkers have been identified to date, the presence of cytokine dysregulation underscores the critical role of neuroinflammation in SMA pathophysiology. Future research should aim to delineate the inflammatory components involved and identify combinations of cytokines better suited for monitoring this highly heterogeneous disease [30].

Studies in pediatric cohorts suggest specific cytokine associations [73,74,125]. In a study by Zhang et al., MCP-1, IL-7, and IL-8 were found to be elevated at baseline in the CSF of patients with SMA type 1 and were correlated with disease severity [73]. In addition, changes in cytokines such as eotaxin, macrophage inflammatory protein-1β (MIP-1β), and IL-10 were associated with response to nusinersen treatment and improvements in motor function [73]. Rybova et al. reported that SMA-PME, which is caused by a deficiency of acid ceramidase, is characterized by increased plasma levels of cytokines such as MCP-1. In untreated mouse models, the levels of these cytokines progressively increased over the course of the disease, while hematopoietic stem cell transplantation restored and stabilized them. MCP-1 could function as a promising biomarker for assessing disease progression and evaluating therapeutic responses in SMA-PME [74]. Furthermore, Allison et al. observed differential changes in CSF levels of C-C motif chemokine ligand 5 (CCL5) and IL-1 receptor antagonist (IL-1ra) before and after nusinersen treatment in SMA patients. Their findings suggest that elevating SMN levels alone may not be sufficient to suppress the pro-inflammatory glial phenotype [75].

The study conducted by Cheng et al. evaluated 15 SMA patients (9 type 2, median age 4 years; 6 type 3, median age 19.5 years), with 33.3% of type 3 patients retaining ambulation [126]. Motor function and CSF inflammatory markers (MCP-1, tumor necrosis factor-α [TNF-α], IL-10) were assessed before, during (63 days), and after (6 months) nusinersen treatment [126]. In SMA type 2, baseline MCP-1 was 333 pg/mL (vs. 296 pg/mL in type 3; *p* = 0.864), while TNF-α was lower (0.447 pg/mL vs. 0.612 pg/mL in type 3; *p* = 0.289) [126]. In both subtypes, MCP-1 increased significantly by 6 months (*p* = 0.016), whereas TNF-α declined (*p* = 0.004 at 63 days; *p* = 0.021 at 6 months) [126]. IL-10 trends were nonsignificant but inversely correlated with baseline HFMSE scores (*p* = 0.033), particularly in type 3 (*p* = 0.045) [126]. Reductions in IL-10 predicted RULM improvements in both subtypes (*p* = 0.007) [126]. SMN2 copy number influenced baseline MCP-1 (*p* = 0.029) but not TNF-α or IL-10 [126]. The findings suggest neuroinflammation influences SMA severity and treatment response, with MCP-1 and TNF-α dynamics potentially reflecting nusinersen’s therapeutic effects. However, another study conducted by Kobayashi et al. analyzed CSF biomarkers in pediatric SMA patients over two years of treatment and found that inflammatory cytokines (TNF-α and interferon-γ [INF-γ]) evaluated by ELISA remained stable with no significant changes [84].

Another study investigated immune profiles in pediatric and adult SMA patients before and after 6 months of nusinersen treatment [125]. In pediatric SMA patients, baseline serum levels of IL-1β, IL-4, IL-6, IL-10, IFN-γ, IL-17A, IL-17F, IL-22, IL-23, IL-31, IL-33, and TNF-α were significantly elevated compared to reference ranges [125]. After treatment, IL-4, IFN-γ, IL-22, IL-23, and IL-33 decreased significantly [125]. Higher baseline IL-23 correlated with worse motor outcomes, while increased post-treatment IL-10 was associated with better HFMSE scores [125]. In adult SMA patients, baseline serum levels of IL-1β, IL-4, IL-6, IL-10, IFN-γ, IL-17A, IL-22, IL-23, IL-31, and IL-33 were elevated compared to healthy controls, with reductions in IL-4, IL-6, IFN-γ, and IL-17A post-treatment [125]. In both groups, CSF cytokines were detectable but unchanged after therapy [125].

These findings suggest that SMA is associated with systemic inflammation, which is partially modulated by nusinersen, with potential implications for disease monitoring and treatment response. Continued investigation is warranted to further dissect the differences in inflammatory components between SMA patients and healthy individuals, and to identify cytokine profiles more appropriately tailored to the clinical heterogeneity of SMA.

### 2.5. Chitotriosidase 1 and Chitinase-3-like Protein 1

Key representatives of CHIT1 and chitinase-3-like protein 1 (CHI3L1), both of which play significant roles in chitin metabolism and have been implicated in various biological processes and diseases, are considered markers of microglial and astrocytic activation [82]. In a study by Freigang et al. [70], CHIT1 concentrations were found to be elevated in the CSF of untreated patients with SMA compared to controls, although CHIT1 levels did not correlate with disease severity. Notably, CHIT1 levels significantly increased following treatment with nusinersen, which may reflect an off-target immune-like response [70]. The study by Freigang et al. analyzed CHIT1 in 58 adult and 21 pediatric SMA patients (types 1–3) with a median age of 31 years, comparing them to an age- and sex-matched control group of 30 individuals (23 healthy, 7 with non-neuroinflammatory conditions) [70]. Andrés-Benito et al. reported that CHIT1 expression was upregulated in microglia and macrophages within the spinal cord of SMA patients, and that CSF levels of CHIT1 were associated with disease severity and progression in adult SMA type 2 and 3, suggesting a role in SMA-related neuroinflammation [65]. De Wel et al. revealed that CHIT1 levels in CSF showed a significant increase (*p* = 0.048) over the 22-month nusinersen treatment period in adult SMA patients [83]. However, unlike CHI3L1, these changes in CHIT1 did not correlate with clinical outcome measures, despite its role as a neuroinflammatory biomarker. This dissociation suggests that while CHIT1 may reflect ongoing biological processes during treatment, it lacks utility as a clinically meaningful biomarker for monitoring therapeutic response in adult SMA [83]. Kobayashi et al. [84] found that CHIT1 levels in CSF showed significant changes during nusinersen treatment in pediatric SMA patients, distinguishing it from stable inflammatory cytokines (TNF-α and INF-γ). Specifically, CHIT1 concentrations decreased significantly between the first and second year of therapy (*p* < 0.05), suggesting its potential role as a dynamic biomarker of treatment response. Additionally, an inverse correlation trend was observed between CHIT1 levels and motor function (HINE-2 scores), implying that declining CHIT1 may reflect clinical improvement. These findings position CHIT1 as a promising CSF biomarker for monitoring therapeutic efficacy in pediatric SMA, warranting further validation in larger cohorts to confirm its predictive value [84].

CHI3L1, also known as YKL-40, produced by various cell types, showed inconsistent changes in CSF and plasma across different studies; while some reports indicated a disease association, its specific role and mechanism in SMA remain unclear [65,83]. For example, CSF levels of CHI3L1 decreased significantly (*p* = 0.037) in adult SMA type 3–4 patients who showed improvements in upper limb function (RULM scores) [83]. Additionally, CHI3L1 exhibited a strong correlation with NF levels in CSF (rho = 0.76), suggesting a potential link between neuroinflammation and neurodegeneration in SMA [83]. However, another study conducted by Andrés-Benito et al. evaluated CHI3L1 in CSF samples from adult SMA type 2 and 3 patients before and during nusinersen treatment [65]. Results showed no consistent changes in CHI3L levels over the course of therapy, suggesting that this marker may not be significantly altered in slowly progressing adult SMA [65]. Unlike more aggressive neurodegenerative conditions, where CHI3L often reflects glial activation and inflammation, its stability in this cohort implies that adult SMA may not involve the same degree of neuroinflammatory activity detectable through this biomarker. The findings align with the overall observation that common neurodegeneration and inflammation markers remained largely unchanged in adult SMA patients following nusinersen treatment, possibly due to the disease’s milder progression compared to pediatric forms.

Studies show divergent CHIT1 patterns in SMA patients: adult levels increase post-nusinersen (reflecting neuroinflammation without clinical correlation), while pediatric levels decrease during treatment (suggesting treatment response), with an inverse trend between CHIT1 and motor function in children. These findings highlight CHIT1’s age-dependent biomarker potential in SMA [70,83,84]. The diagnostic and prognostic utility of CHIT1 and CHI3L1 in SMA thus warrants further investigation [65].

### 2.6. Serum and Urinary Creatinine

Xing et al. [30] reviewed previous studies and proposed that in neuromuscular disorders, reduced Crn values and low serum Crn concentrations are associated with increased disease severity. In SMA, serum Crn levels are negatively correlated with both disease severity and genetic instability, suggesting its potential as a prognostic biomarker. This correlation has been observed across different age groups. In terms of treatment response, serum Crn shows limited sensitivity to most therapies. During nusinersen treatment, changes in Crn concentrations vary among different SMA subtypes. Some studies report slight increases in type 2/3 SMA patients [79] and decreases in type 1 SMA patients [127] during nusinersen treatment, suggesting variability in treatment response based on disease severity. Moreover, the underlying pathophysiology of SMA and the nephrotoxicity induced by gene therapy may interfere with the clinical interpretation of Crn levels [30].

Furthermore, a retrospective study by Bahadır Şenol et al., involving 33 SMA patients, found that baseline serum Crn levels were higher in type 2 patients compared to type 1. However, during the administration of nusinersen, there were no statistically significant alterations in serum Crn levels observed in either group, indicating that nusinersen has no apparent adverse effect on serum Crn in SMA patients [80]. Type 1 patients showed a significant increase in urine Crn following nusinersen loading doses, correlating with motor function improvements [80]. The findings suggest that urine Crn may serve as a potential biomarker for treatment response in SMA Type 1, reflecting functional improvements without compromising renal safety [80]. In a study of 46 pediatric patients with 5q SMA types 1–3 treated with nusinersen, baseline data revealed low Crn levels in 31 patients [128]. During treatment, one SMA type 3 patient exhibited transiently elevated Crn levels at the third (V3) and fourth (V4) injections (52.6 and 71.7 µmol/L, respectively), which normalized by the fifth follow-up (V5) without intervention [128]. Overall, Crn levels did not show significant changes across the cohort during nusinersen treatment (*p* > 0.05) [128]. However, regression analysis indicated a significant increase in urea levels (*p* = 0.017) and a significant decrease in cystatin C levels (*p* = 0.000) per injection, suggesting potential kidney-related metabolic changes despite stable Crn measurements [128]. These findings highlight variability in individual responses, with most patients maintaining low creatinine levels, consistent with SMA-related muscle loss, while a subset may experience transient fluctuations [128]. Another study analyzed retrospective data from 58 SMA 5q patients (including 21 pediatric cases) to assess the clinical relevance of biomarkers, including Crn. Results showed no significant subgroup differences in serum Crn levels across SMA subtypes, suggesting that Crn alone is not a reliable marker for disease monitoring or progression in SMA [129].

In a study by Chen et al. [76] on Chinese adult SMA patients receiving nusinersen, serum Crn levels showed a certain degree of correlation with motor function and activities of daily living. However, over the 18-month treatment period, changes in serum Crn were not consistently aligned with the dynamic changes in functional scores. By contrast, the creatinine-to-cystatin C ratio (CCR) demonstrated better performance in reflecting treatment-associated changes in motor function and may serve as a more effective monitoring biomarker [76]. In adolescents and adults with SMA, serum Crn levels are significantly reduced and closely related to disease severity [81]. After adjusting for age, sex, and body mass index (BMI), serum Crn levels showed strong positive correlations with multiple motor function indicators, explaining up to 83.5% of the variance in motor performance. This finding supports its potential use as a biomarker for evaluating disease severity in SMA. However, larger cohort studies and longer follow-up periods are needed to validate its clinical utility [81]. In addition, in a preliminary study conducted by Li et al., involving 10 adult SMA patients and 10 patients with ALS, serum Crn levels were significantly lower in SMA patients than in amyotrophic lateral sclerosis (ALS) patients. Furthermore, serum Crn in SMA patients was correlated with baseline Amyotrophic Lateral Sclerosis Functional Rating Scale—Revised (ALSFRS-R) scores, disease progression rate, and pulmonary function, suggesting its potential utility as a biomarker of other motor neuron diseases [124].

Collectively, while correlations have been observed across various age groups, the most substantial evidence supporting the potential of serum Crn as a biomarker for disease severity primarily stems from studies involving adult and adolescent patients with SMA. Urinary Crn, on the other hand, shows promise as a potential biomarker for assessing treatment response in SMA Type 1. However, further dedicated studies are needed to specifically evaluate serum Crn as a biomarker in pediatric SMA patients, especially in infants and young children.

### 2.7. Creatine Kinase

CK is a biomarker indicative of muscle damage. Its reference range varies depending on sex, laboratory standards, and assay methods. CK levels are influenced by factors such as age, sex, and muscle mass, and do not correlate precisely with the actual extent of muscle injury. Elevations in CK can result from a wide range of causes—physiological, exercise-related, pathological, or pharmacological—highlighting its importance in the diagnosis and evaluation of muscle-related diseases and drug-induced side effects [77].

Several studies have reported elevated serum CK levels in patients with SMA [78,79]. In a multicenter observational study of 206 adult patients with 5q-associated SMA, Freigang et al. [79] found a significant correlation between serum CK levels and disease severity. Moreover, during an 18-month treatment course with nusinersen, CK levels showed a significant decline, suggesting that CK may serve as a potential biomarker for both disease severity and treatment response in adult patients with SMA. The dynamic reduction in CK levels could reflect reduced muscle damage and improved energy metabolism following treatment [79]. A recent study on Chinese adult SMA patients receiving nusinersen reported that serum CK levels were negatively correlated with motor function scores, including the HFMSE, RULM), and Barthel Index (BI). However, no significant changes in log_10_-transformed CK levels were observed over the 18-month treatment period, and its utility for monitoring motor function changes was inferior to that of the CCR [76].

It is also important to note that various factors can influence CK levels, including physiological conditions, exercise and trauma, endocrine disorders, genetic diseases, and medications such as adrenergic stimulants and lipid-lowering agents [77]. Therefore, when interpreting serum CK levels, these confounding factors should be carefully considered and excluded. The current evidence supporting the utility of CK as a biomarker for SMA primarily derives from studies in adult patients. Dedicated investigations evaluating CK as a biomarker specifically in pediatric SMA populations are limited.

### 2.8. Glial Fibrillary Acidic Protein

GFAP is an intermediate filament protein classified as type III, predominantly expressed in astrocytes within the brain. It serves as a surrogate marker of astrocyte activation and the subsequent neuroinflammatory response [85,86,130]. Under normal physiological conditions, GFAP expression increases gradually with age, and this age-related upregulation may be closely associated with brain injury. In recent years, GFAP has garnered increasing interest for its potential utility in CNS disorders, and its role has also been explored in the context of SMA [30]. In a study by Olsson et al. [48], CSF GFAP levels were significantly elevated at baseline in pediatric SMA patients compared to controls. Following treatment with nusinersen, GFAP levels declined, although no correlation was observed between changes in GFAP and improvements in motor function. Elevated baseline levels of GFAP were interpreted as evidence of astrocytosis in SMA, while post-treatment reductions suggested that nusinersen might attenuate glial activation [48]. A study containing 58 adult patients and 21 children, conducted by Freigang et al. [51], also reported that although CSF GFAP levels did not differ significantly between SMA patients and age- and sex-matched controls, they were associated with disease severity. In some patients, GFAP levels decreased following nusinersen treatment [51].

Taken together, while GFAP concentrations in CSF show some association with disease severity in SMA, particularly in younger patients with more severe forms (e.g., type 1), they do not serve as a reliable diagnostic or prognostic biomarker, especially in adults or milder SMA subtypes (type 2/3) [48,51]. The limited changes in GFAP levels following nusinersen treatment and the lack of correlation with motor function improvements suggest that glial activation, though potentially involved in SMA pathology, may not be a primary driver of disease progression or treatment response [48,51]. Further research is needed to clarify the role of GFAP in SMA and its utility in monitoring therapeutic outcomes, particularly across different age groups and disease subtypes.

### 2.9. Amyloid-β 40 and 42

Aβ peptides 40 and 42 are expressed in various cell types and are derived from the proteolytic cleavage of amyloid precursor protein (APP). These peptides circulate in human body fluids. Aβ40 and Aβ42 are the principal components of amyloid plaques in AD and are highly neurotoxic due to their abnormal deposition and accumulation in the extracellular matrix. Consequently, they are frequently investigated as potential biomarkers for AD [53]. Additionally, β-site APP cleaving enzyme-1 (BACE1), which serves as the rate-limiting enzyme in the production of Aβ, has garnered considerable interest due to its potential as a therapeutic target in AD [54].

In a study conducted by Jessika Johannsen et al. involving patients with Type 1 SMA, a significant correlation was observed between BACE1 levels measured at day 180 and subsequent changes in motor function outcomes evaluated at day 300 [55]. Importantly, the overall concentration of BACE1 remained stable and did not show any significant fluctuations throughout the duration of the study [55]. Verma et al. showed that in nine younger and three older children SMA patients treated with nusinersen, levels of soluble APP α (0.055/mo, 95% CI 0.016–0.099, *p* = 0.012) and soluble APP β (0.054/mo, 95% CI 0.034–0.075, *p* < 0.001) in CSF demonstrated statistically significant decreases regardless of age [30,56]. A single-center study provides preliminary evidence that CSF Aβ42 levels increased in adults with SMA types 2 and 3 following nusinersen treatment, with sustained elevations observed at day 180 (*p* = 0.012) and day 420 (*p* = 0.018) compared to baseline [57]. These findings suggest a potential association between Aβ expression and SMA and support the hypothesis that nusinersen may help maintain and enhance the viability of surviving motor neurons [30,56,57]. While nusinersen treatment led to a reduction in soluble APPβ levels in adult SMA patients, this change did not correlate with clinical outcomes, suggesting that sAPPβ may not be a reliable biomarker for monitoring treatment response [65]. Introna et al. revealed that Aβ40 levels in CSF were not significantly changed following nusinersen treatment [57]. Given the small sample size and lack of a control group, further large-scale studies are needed to validate Aβ40 and Aβ42 as reliable biomarkers of treatment response in SMA [57]. Moreover, the study conducted by Walter et al. indicated that the levels of Aβ40 and Aβ42 in the CSF of adult patients diagnosed with 5q-associated SMA type 3 remained relatively stable and did not show a clear trend of change in response to treatment [58]. The discrepancies in the above studies could stem from differences in SMA subtypes, disease duration, or patient age, emphasizing the need for larger, controlled longitudinal studies to clarify the utility of biomarkers. Current evidence supports further investigation into Aβ42 and BACE1 as potential indicators of neuronal health, but their role in clinical monitoring remains uncertain.

### 2.10. Tau Protein

Tau is a multifunctional protein expressed in neurons and other cell types, where it plays critical roles in maintaining cellular physiology. Aberrant modifications and aggregation of tau are implicated not only in a variety of neurodegenerative disorders, such as AD, but also in cardiovascular diseases, cancer, and other pathological conditions. In these diseases, tau influences cellular function through interactions with a wide array of proteins [60,61]. Multiple studies have reported that nusinersen treatment can yield clinical benefits in SMA, particularly in improving motor function in some patients. However, findings related to biomarkers remain inconsistent. While some reports suggest that total tau and related proteins may serve as indicators of therapeutic response or disease status, others—particularly in adult SMA populations—have found no clear correlation between certain biomarkers, treatment, or motor function scores. Additionally, nusinersen therapy may induce changes in the blood–brain barrier (BBB) and immune-related processes.

Pediatric SMA Type 1 shows markedly elevated baseline tau levels (939 ± 159 pg/mL) that significantly decrease with treatment (*p* = 0.01), with this reduction strongly correlating with motor function improvement in twelve children with SMA (rho = −0.85, *p* = 0.0008) (Olsson et al. [48]). Johannsen et al. [55] observed a significant inverse correlation between CHOP INTEND scores and tau concentration, specifically in SMA type 1 patients following nusinersen treatment, while no such correlation was found with RULM and HFMSE scores in types 2 and 3 (Johannsen et al. [55]). Although tau levels decreased across all SMA patients during treatment, statistically significant reductions were only detected in type 1 patients at 2 months post-treatment and in type 2 patients after 10 months (Johannsen et al. [55]). However, based on findings from Olsson et al. [48], NfL emerges as a superior biomarker to tau for monitoring nusinersen treatment response in SMA patients, particularly for reasons of greater magnitude of change, with NfL showing a substantially larger decrease (4598 ± 981 to <380 pg/mL) compared to tau, and the rate of NfL reduction (−879.5 pg/mL/dose) that was nearly 8-fold greater than tau (−112.6 pg/mL/dose) (Olsson et al. [48]). The small study population limits generalizability and may have obscured age-dependent treatment effects. Larger cohort studies are required for further validation.

So far, various studies have shown a complex picture regarding the tau protein’s role as a biomarker for adult SMA patients (types 2 and 3) undergoing nusinersen treatment. First, baseline CSF tau levels show conflicting results across studies, with some reporting normal levels (<290 pg/mL) in type 3 patients (Totzeck et al. [59]), while others document mild but significant elevation (181.29 vs. 176.21 pg/mL) in untreated type 2–3 patients (Andrés-Benito et al. [65]). Treatment response patterns are also inconsistent. For example, most studies found stable CSF tau levels throughout treatment (2–22 months) (Walter et al. [58], Milella et al. [63], Andrés-Benito et al. [65]). Walter et al. [58], reported phospho-tau reduction, suggesting possible axonal protection. No correlation between tau and motor function improvements was established. Plasma and CSF tau levels showed a discrepancy, with significantly lower plasma tau in SMA patients (2.98 vs. 7.51 pg/mL) (Andrés-Benito et al. [65]), highlighting potential compartment-specific tau metabolism in SMA. Current evidence does not support the routine use of this biomarker for treatment monitoring in adult SMA patients. The field requires more sophisticated approaches to capture the complexity of adult SMA pathophysiology. The variability in adult populations highlights the need for age-specific reference ranges, standardized sampling protocols, and further research on the mechanisms underlying tau dynamics in SMA. The consistent findings across multiple studies support the incorporation of CSF tau monitoring into clinical practice for pediatric SMA management, while underscoring the need for additional investigation in adult populations [48,55,58,59,62,63,64,65].

### 2.11. S100 Calcium-Binding Protein B

S100B is predominantly localized in astrocytes of the CNS and is considered a reliable biomarker of neural injury [87,88]. To date, research on S100B levels in patients with SMA remains limited. In a study by Totzeck et al., blood and CSF samples were collected from 11 adult patients with SMA type 3 during the loading phase of nusinersen treatment [59]. Mild elevations of S100B were detected in the CSF or serum of some patients; however, these changes were non-specific and unrelated to the course of treatment [59]. Similarly, a study by Šimić et al. [62] analyzed the relationship between CSF S100B levels, nusinersen dosage, and motor function assessment scores in 30 patients with SMA types 1 to 3. Their findings showed no significant difference in S100B levels compared to baseline following various dosing regimens, and no correlation was found between S100B concentrations and either nusinersen dosage or motor function scores [62]. These findings suggest that S100B may not serve as a reliable biomarker for monitoring therapeutic response in SMA patients receiving nusinersen treatment.

### 2.12. Conventional Cerebrospinal Fluid Parameters

CSF contains specific cellular and chemical components and is extensively used for the diagnostic evaluation of CNS diseases, monitoring therapeutic efficacy, and predicting patient prognosis. However, several studies have reported that conventional CSF parameters in patients with SMA are generally comparable to those in control subjects, thereby limiting their utility as diagnostic or prognostic biomarkers for SMA [63,66,67,122].

Increases in CSF cell counts, total protein levels, and the albumin quotient (QAlb) are generally interpreted as markers of non-specific inflammatory processes [68,69]. Some studies have shown that in SMA patients receiving nusinersen treatment, the cellular composition of CSF remains remarkably stable, even though there is a mild increase in total protein levels—still within the normal reference range [66,67,70,71,72,122]. Hegen and McCudden noted that elevations in CSF cell counts, protein concentration, and QAlb typically reflect nonspecific inflammatory progression [68,69]. These findings were also echoed in studies reporting a mild increase in QAlb following nusinersen treatment [50,70,71,72,122]. Some investigators attribute these mild inflammatory changes not to the pharmacological action of nusinersen itself, but rather to repeated intrathecal injections. Gingele et al. [66] observed the presence of macrophages containing purple and blue cytoplasmic granules—termed “nusinophages”—in the CSF of SMA patients treated with nusinersen, beginning from the second lumbar puncture. These cells, which were absent at the initial puncture, accounted for 0.5–6.5% of total white blood cells and remained relatively stable over time. This phenomenon has been reported in studies involving patients of differing ages, but specific differences between pediatric and adult populations regarding nusinophages have not been described. Importantly, these specific cell types have not been observed in the CSF of patients suffering from other neurological conditions. While the origin and significance of these unique phagocytes remain unclear, it is hypothesized that the intracellular granules may contain nusinersen. Further investigation is needed to determine their potential impact on disease progression or treatment response [66].

Wurster et al. [122] and Milella et al. [63] also examined other CSF indicators—including oligoclonal bands, lactate levels, markers of axonal injury (e.g., neurofilament light and phosphorylated heavy chains), CSF/serum glucose ratio, and QAlb—but found no consistent associations with disease state or treatment response in adolescent and adult SMA type 2 and 3 patients. Overall, these findings underscore the need for further research to clarify the clinical relevance of conventional CSF parameters in SMA and their potential as biomarkers.

### 2.13. Gemin Proteins

The SMN complex orchestrates the cytoplasmic assembly of snRNPs by mediating the association of Sm protein rings with pre-snRNAs at conserved Sm sites [102,103]. Disruption of this process impairs spliceosome function, contributing to the pathogenesis of SMA and cancer [102,103]. Structural studies reveal specialized roles for Gemin proteins. For example, Gemin2 regulates snRNP core assembly through negative cooperativity, maintaining Sm proteins in a closed conformation for RNA selectivity before RNA-induced release [104]. The loss of Gemin2 is embryonically lethal in vertebrates and causes motor neuron defects in model organisms [104]. Gemin5 ensures assembly fidelity through dual recognition of Sm sites and m^7^G caps via its WD40 domain [102]. SMN1 mutations reduce SMN protein levels, destabilizing the SMN-Gemin complex and triggering aberrant Gemin degradation, thus disrupting minor spliceosome function, potentially explaining the vulnerability of motor neurons [103]. Organoid models reveal that SMN deficiency also alters the fate of neuro-mesodermal progenitors, suggesting that combined developmental and splicing defects drive the disease [103]. Gemin complex stability (e.g., spinal Gemin2 levels) and its snRNP assembly efficiency correlate with the extent of splicing dysfunction in SMA disease models. When combined with minor intron retention levels, these metrics may provide a dynamic biomarker panel to track disease progression and evaluate the efficacy of SMN-targeted therapies (e.g., ASOs) [131,132].

Despite their mechanistic importance, Gemin proteins face hurdles as standalone SMA biomarkers, including low specificity, detection barriers, and a lack of clinical correlation. Gemin proteins (e.g., Gemin2, Gemin5) are integral to the SMN complex and snRNP assembly, but their dysfunction is not exclusive to SMA [105]. For instance, Gemin5 mutations are associated with neurodevelopmental disorders like autism and intellectual disability, complicating their specificity for SMA diagnosis [105]. Current SMA diagnostics rely on *SMN1* genotyping and SMN protein quantification, which are more directly tied to disease mechanisms [103]. While SMN protein restoration via therapies like nusinersen or Zolgensma shows clear clinical benefits, no studies have established a direct relationship between Gemin protein dynamics (e.g., Gemin2 levels) and SMA severity or therapeutic response. Gemin proteins are primarily intracellular (e.g., in Cajal bodies), making their detection in blood or CSF challenging [102,104]. Current methods (e.g., immunoprecipitation, mass spectrometry) require highly specialized protocols. Unlike SMN protein ELISA or *SMN2* copy number analysis, no clinically validated assays exist for Gemin proteins in SMA patients. Most data come from cell or animal models (e.g., zebrafish, *Drosophila*), limiting translational relevance.

Targeting the Gemin complex presents novel therapeutic avenues. One strategy involves enhancing the RNA-binding affinity of Gemin5 to boost spliceosomal accuracy [107,133]. Another approach focuses on inhibiting detrimental interactors, such as pICln and Tgs1, to restore proper snRNP biogenesis [107,133]. These strategies could synergize with current SMN-enhancing treatments, potentially tackling both splicing errors and developmental abnormalities associated with the deficit [107,133]. Future directions include the advancement of ultrasensitive analytical techniques, such as SIMOA, for the quantification of trace levels of Gemin proteins in various biofluids [134]. Additionally, investigating composite biomarker panels that integrate Gemin proteins with other potential markers and validating therapeutic targets using patient-derived models are potential strategies for further investigation.

## 3. Non-Molecular Biomarkers for Spinal Muscular Atrophy

### 3.1. Electrophysiological Biomarkers for Spinal Muscular Atrophy

CMAP, measured via nerve conduction studies, is typically assessed through parameters such as amplitude, latency, and F-wave responses. MUNE utilizes electrophysiological techniques—including nerve stimulation, electromyographic (EMG) signal decomposition, and surface EMG interference pattern analysis—combined with CMAP and EMG signals at varying contraction levels, to estimate the number of motor units. Commonly used methods include the MUNIX and the multi-point incremental stimulation technique [108].

Numerous studies have demonstrated that CMAP and MUNE can be used to monitor the health, integrity, and function of peripheral motor units in early-stage patients with SMA [41,109,110,111,112,113]. In SMA models, parameters derived from electrical impedance myography (EIM) have been shown to correlate with various clinical indicators and can reflect both treatment efficacy and disease status. As a non-invasive approach, EIM holds promise as a pediatric biomarker [114]. When combined with machine learning algorithms, EIM shows strong potential for detecting infantile-onset SMA and monitoring disease progression. Linear discriminant analysis (LDA) models have demonstrated high classification accuracy and effective tracking capabilities. However, this technique still faces limitations, including a lack of correlation with other clinical measures, extensive data filtering requirements, and discontinuation of specific EIM devices. Further research is needed to address these challenges and validate its clinical utility [115].

Electrophysiological biomarkers in SMA demonstrate distinct age- and subtype-specific patterns. In pediatric SMA, CMAP shows dynamic changes—rapid initial decline in type 1 infants, followed by stabilization, modest progressive decline in type 2 children, and relative stability in type 3 adults [41,109,112]. CMAP serves as a valuable marker for disease onset, progression, and treatment response in children, while MUNE is significantly reduced in symptomatic pediatric cases. For adults, the Motor Unit Number Index (MUNIX) of hand muscles emerges as a promising biomarker, though it requires further validation. While NFs remain robust biomarkers in children, their adult utility is limited by age-related variability, leading to preferential use of CMAP in older patients. EIM correlates well with muscle strength in children but lacks sufficient adult data. Persistent neuromuscular junction defects detected by repetitive nerve stimulation in treatment-unresponsive adults highlight its potential for assessing junctional dysfunction. These findings collectively underscore the age-dependent utility of electrophysiological markers, with CMAP and MUNE being most informative in pediatric populations, while MUNIX shows growing promise for adult assessment.

Barp et al. investigated facial nerve involvement in 21 SMA type 2 patients, 16 type 3 patients, and 27 healthy controls using CMAP and MUNIX of the orbicularis oculi muscle [135]. The results demonstrated significantly reduced CMAP amplitudes and MUNIX scores in SMA patients compared to healthy controls (*p* < 0.0001), confirming neurophysiological evidence of facial motor unit loss. Notably, SMA 3 patients exhibited higher CMAP and MUNIX values than SMA 2 patients, reflecting less severe cranial nerve impairment in milder SMA subtypes. Both techniques proved feasible and well-tolerated, with CMAP and MUNIX effectively discriminating between SMA subtypes and quantifying facial motor unit loss. However, no significant differences were observed based on functional status or nusinersen treatment, suggesting these measures primarily reflect disease subtype rather than treatment response. The findings establish CMAP and MUNIX as reliable tools for assessing cranial nerve vulnerability in SMA [135]. Another study demonstrated the clinical utility of MUNIX and CMAP as electrophysiological biomarkers in pediatric SMA (type 2) patients [136]. Compared to matched controls, patients showed significantly reduced MUNIX scores and CMAP amplitudes (*p* < 0.05) along with increased MUSIX values in both abductor digiti minimi (ADM) and abductor pollicis brevis (APB) muscles, indicating substantial motor unit loss and compensatory reinnervation in patients [136]. The technique exhibited excellent reliability, with high intraclass correlation coefficients (0.83–0.98) for all parameters. Importantly, CMAP amplitudes correlated with grasp/pinch strength (*p* < 0.05), while MUNIX values showed significant associations with overall motor function (Motor Function Measurement [MFM] scores). These findings validate MUNIX as a sensitive, quantitative measure of motor neuron loss that correlates with clinical measures of disease severity, suggesting its potential utility for monitoring disease progression in pediatric SMA populations patients [136].

### 3.2. Imaging Technologies for Spinal Muscular Atrophy

Ultrasound imaging plays a valuable role in SMA by detecting muscle EI, MT, and the characteristic “moth-eaten” echotexture. Neuromuscular ultrasound in SMA reveals distinct pathological signatures: marked muscle thinning reflecting severe atrophy and significantly elevated echogenicity with a heterogeneous “*moth-eaten*” pattern arising from alternating regions of denervated atrophy and hypertrophic regenerating fibers [116]. Longitudinal studies highlight its dynamic utility, as rapid progression (e.g., quadriceps atrophy and escalating echogenicity within 2–4 months in SMA type 1 infants) correlates with severe phenotypes, underscoring its potential as a biomarker for monitoring disease trajectory and therapeutic response [137]. A recent study conducted by Li et al. demonstrated that ultrasound imaging effectively detects muscle abnormalities in children with SMA type 2/3 [138]. Compared to healthy controls, SMA patients showed significantly reduced MT and shear wave velocity (SWV) in both biceps brachii and quadriceps femoris muscles (*p* < 0.05), indicating pronounced muscle atrophy and altered tissue elasticity. Importantly, quadriceps femoris-MT and quadriceps femoris-SWV exhibited strong positive correlations with HFMSE motor function scores (r = 0.802 and r = 0.56, respectively, *p* < 0.001), suggesting these ultrasound parameters reliably reflect disease severity [138]. The findings establish conventional ultrasound as a valuable, non-invasive tool for quantifying SMA-related muscle degeneration and monitoring clinical progression in pediatric patients [138]. Moreover, Pelosi et al. demonstrated that muscle ultrasound is an effective method for detecting significant structural abnormalities in adults with SMA compared to matched controls [139]. SMA patients exhibited markedly reduced mean muscle thickness (1.3 vs. 1.9 cm, *p* < 0.05), increased muscle echogenicity (106 vs. 67 grayscale units), and greater subcutaneous thickness (0.9 vs. 0.3 cm), along with altered M:S ratios for both thickness and echogenicity [139]. Notably, these ultrasound parameters correlated with disease severity, showing the most pronounced changes in nonambulatory patients compared to ambulatory individuals [139]. The findings suggest that muscle ultrasound could serve as a valuable biomarker for quantifying disease progression in adult SMA, though larger validation studies are needed to confirm its clinical utility [139].

A prospective study of 150 children with suspected neuromuscular disorders revealed that quantitative muscle ultrasound—assessing echo EI and thickness in four muscles—achieves 71% sensitivity and 91% specificity for diagnosis using defined EI thresholds (>3.5 standard deviation [SD) in one muscle, >2.5 SD in two, or >1.5 SD in three muscles), with a high positive predictive value (91% [82–98% CI]). Normal EI (<2.0 SD in all muscles) excludes disease with 91% sensitivity (negative predictive value [NPV) 86% [76–96% CI]), though specificity declines to 67% [116,140]. Notably, sensitivity drops to 75% in children under 3 years, with a 25% false-negative rate encompassing early-stage SMA, Duchenne muscular dystrophy, and metabolic myopathies, reflecting delayed structural manifestation in infantile-onset disease [116,140]. Therefore, early-stage SMA in infants (<3 years) may exhibit false-negative findings due to initially subtle structural changes, necessitating cautious interpretation in young cohorts [116,140]. Overall, these quantitative parameters correlate with disease severity and functional status, allowing for the sensitive detection of early neuromuscular pathology. Ultrasound provides a non-invasive modality for clinical diagnosis, disease monitoring, and assessment of treatment response, while highlighting the need for age-adjusted interpretation, which may lead to undetected conditions [116,137,140].

MSOT reveals distinct signatures in pediatric SMA patients [117]. Muscle tissue demonstrates a heterogeneous “moth-eaten” optoacoustic signal pattern, characterized by alternating patchy areas of high and low signal intensity, contrasting sharply with the homogeneous signal band observed in healthy volunteers [117]. Quantitative analysis shows significantly lower mean signal intensity at 800 nm (hemoglobin absorption peak) in SMA patients compared to controls (*p* = 0.0082, single-wavelength measurement), with signal intensity at this wavelength positively correlating with clinical motor function scores assessed by HFMSE and RULM (r = 0.63–0.72, *p* < 0.01) [117]. Additionally, SMA patients exhibit elevated peak signal intensity across multiple wavelengths (680–850 nm) and higher maximum lipid signal than controls (*p* = 0.0050), suggesting possible fat infiltration [117]. These findings demonstrate MSOT’s capability for visualization of muscle degeneration, representing a novel, non-invasive, and quantitative imaging technique for evaluating disease severity, tracking progression, and assessing therapeutic response in pediatric SMA [117]. While MSOT shows promise for pediatric SMA assessment, its potential as a biomarker in adult SMA patients requires further investigation to establish its clinical utility across different age groups and disease stages.

These imaging modalities provide non-invasive approaches for monitoring and evaluating SMA and demonstrate high sensitivity in identifying disease severity, making them promising candidates as emerging biomarkers.

## 4. Emerging Technologies in Biomarker Discovery for Spinal Muscle Atrophy

Previous studies utilizing spatial transcriptomics in SMA have demonstrated that high-throughput omics approaches can facilitate the identification of potential biomarkers for SMA. For example, plasma cathepsin D levels showed a non-significant downward trend in infants with SMA compared to age-matched healthy controls in baseline measurements from the NeuroNEXT biomarker study (*p* > 0.05) [141]. Schorling et al. identified cathepsin D as a promising biomarker in SMA through untargeted proteomic CSF analysis of patients treated with nusinersen [142]. Validation in 31 pediatric SMA patients (types 1–3 and presymptomatic) revealed significantly decreased cathepsin D levels in treatment-responsive patients aged ≥2 months at therapy initiation, with this decline being most pronounced in responders [142]. The downregulation was also observed in SMA muscle biopsies. While cathepsin D showed potential across all age groups, its strongest biomarker utility emerged in older patients, either in combination with neurofilament light chain in adolescents or as a standalone marker in adults. These findings position cathepsin D as a candidate biomarker for monitoring treatment response, particularly in older SMA type 1 patients, where therapeutic assessment remains challenging [142]. At the genetic level, prior research has reported downregulation of key genes involved in the “actin cytoskeleton regulation” pathway, such as Rho-associated coiled-coil containing protein kinase 1 (ROCK1), Ras homolog family member A (RHOA), and actin beta (ACTB), in the same peripheral samples, alongside upregulation of heat shock protein family A member 7 (HSPA7) in serum [30].

Integrated transcriptomic analysis of 39 human SMA microarray datasets, employing weighted correlation network analysis (WGCNA), gene set enrichment analysis (GSEA), and Cytoscape network analysis (v3.10.3), has identified disease severity-associated gene modules and revealed a TNFα-mediated core regulatory network with three principal downstream signaling axes [143,144]. First, the TNF-α-bone morphogenetic protein 4 (Bmp4)-Serpine1-GATA binding protein 6 (Gata6) axis is implicated in neurodevelopmental and cardiogenesis processes [143]. In severe SMA mouse models, TNF-α upregulation suppresses Bmp4 while enhancing Serpine1 and Gata6 expression, disrupting neuronal and cardiomyocyte differentiation. Second, the TNF-α-prostaglandin-endoperoxide synthase 2 (Ptgs2)-B-cell lymphoma 2 (Bcl2) axis is associated with skeletal system development. Elevated Ptgs2 and Bcl2 promote bone resorption and inhibit osteoblast differentiation, contributing to SMA-associated osteopenia [143]. Third, the TNF-α-IL-6-contactin 1 (CNTN1) axis is involved in nervous system development [143]. IL-6, acting as a myokine, upregulates Cntn1 expression, potentially mediating neuroprotective compensatory mechanisms. These findings establish key targets (Bmp4, Serpine1, Ptgs2, Gata6) as central to SMA’s multisystem pathology, with TNF-α hyperactivation as the critical upstream driver [143]. Therapeutic strategies targeting TNF-α or its downstream effectors (e.g., Bmp4 overexpression or Serpine1/Ptgs2 inhibition) may ameliorate neurodevelopmental, cardiac, and skeletal impairments. Notably, the expression profiles of these targets correlate with disease severity in murine models, providing experimental validation [143,144].

Complementary work by Nichterwitz et al. delineated motor neuron subtype-specific responses through Gene Ontology (GO) analysis [134]. Vulnerable somatic motor neurons activate TRP53-mediated apoptotic signaling and p53-class mediator transduction (involving Cdkn1a, Pmaip1), coupled with RNA processing defects (e.g., Snrpa1 dysregulation)—responses absent in spared rubrospinal and visceral motor neurons [134]. Resistant ocular motor neurons uniquely upregulate neuroprotective pathways: Neurotransmitter release regulation (synaptotagmin [Syt]1, Syt5, complexin 2 [Cplx2]), Pro-survival signaling (growth differentiation factor 15 [GDF15], cell adhesion molecule L1-like protein [Chl1], leukemia inhibitory factor [Lif]), Oxidative stress defense (Aldh4a1), and Anti-apoptotic mechanisms (Pak4) [134]. Functional studies identified GDF15 as a key protector, enhancing human spinal motor neuron survival via phosphoinositide 3-kinase/protein kinase B (PI3K/AKT) activation, while Syt1 upregulation may compensate for synaptic deficits [134]. These findings suggest that augmenting ocular motor neuron-protective pathways (e.g., GDF15/Syt1 modulation) could lead to targeted therapeutic strategies for mitigating motor neuron degeneration in SMA [134]. However, these findings were derived from murine models, highlighting the need for further validation in human studies.

In a more recent study, Lu et al. employed spatial transcriptomics combined with multiplex immunohistochemistry and discovered that, in the brains of adult patients with SMA type 1–3, degenerating neurons were surrounded by IL-18–expressing myeloid cells and clusters of cytotoxic CD8⁺ T cells. These findings suggest that myeloid cell–induced recruitment of cytotoxic immune cells may contribute to the pathogenesis of SMA [145]. Liguori et al. [22] conducted a 10-month transcriptomic analysis of adult SMA patients receiving nusinersen. Their findings revealed that patients who showed clinical improvement exhibited an increased *SMN2/SMN1* expression ratio in adult patients with SMA types 2–4, whereas those with stable disease had a decreased ratio. Additionally, 38 differentially expressed genes (e.g., TNF Receptor-Associated Death Domain [TRADD], JunD Proto-Oncogene [JUND]) returned to baseline levels following treatment, and dysregulation of several microRNAs, including miR-146a-5p, was observed. Notably, miR-146a-5p targets *SMN1*, and the data implicated the involvement of NOTCH, nuclear factor kappa B (NF-κB), and Toll-like receptor signaling pathways in SMA pathophysiology [22]. These studies underscore the value of omics-based approaches in the search for novel biomarkers in SMA. However, the process of identifying optimal biomarker candidates from large-scale datasets remains a considerable challenge. Further research is warranted to validate previously identified markers and to uncover new, clinically relevant biomarkers.

Epigenetic research in SMA has led to the establishment of a multidimensional biomarker system. Research indicates that the methylation patterns of the *SMN2* gene play a critical role in its expression and subsequent implications for SMA. Specific methylation differences in key regulatory genes (such as *SLC23A2*, *NCOR2*, and *DYNC1H1*), and profiles of non-coding RNAs (including serum myomiRs and circRNAs) hold significant value for stratifying disease severity, monitoring therapeutic responses, and refining clinical subtyping [146]. Notably, epigenetic regulation contributes to early neurodevelopmental defects by influencing the fate determination of neuromesodermal progenitor cells. The associated biomarker system encompasses molecular markers (e.g., SOX2, TBXT, ISL1), single-cell transcriptomic features (e.g., WNT pathway gene expression profiles), and tissue morphometric parameters (e.g., neuro-mesodermal region ratios), offering significant insights into the developmental pathways leading to SMA, which will aid in the formulation of targeted epigenetic intervention strategies [103].

In the development of biomarker technologies, the methylation modification of the *SMN1* gene locus (chr5:70239954–70249165) exhibits a characteristic gradient pattern—0–15% in patients, 50–70% in carriers, and 98–100% in non-carriers. When combined with long-read sequencing for allele-level deconvolution of *SMN1/SMN2*, this enables precise diagnosis of SMA and reliable identification of carrier status by detecting regional methylation patterns and *SMN1*-specific read differences [147]. Therapeutic strategy research has revealed that the type I protein arginine methyltransferase (PRMT) inhibitor MS023 can enhance full-length SMN protein expression by modulating the methylation state of the HNRNPA1 protein—shifting from asymmetric dimethylarginine (ADMA) to monomethylarginine/symmetric dimethylarginine (MMA/SDMA)—thereby reducing its binding affinity to SMN2 pre-mRNA and promoting exon 7 inclusion. Combined treatment with MS023 and nusinersen has shown synergistic effects in ameliorating SMA phenotypes and prolonging survival in mouse models. This effect involves transcriptomic correction of neuroinflammatory pathways and dynamic monitoring of HNRNPA1 methylation status, offering a novel direction for epigenetically targeted therapies [148].

Furthermore, significant progress has been made in understanding the epigenetic regulation of the plastin 3 (PLS3) gene, including tissue-specific escape from X-chromosome inactivation, a high copy number of the DXZ4 macrosatellite at Xq23 (≥70 repeat alleles in females), and transcriptional activation of the PLS3 promoter by the chromatin remodeling factor Chromodomain helicase DNA binding protein 4 (CHD4). These elements collectively form a biomarker system for assessing protective effects against SMA and provide new targets for screening asymptomatic individuals with *SMN1* deletion, as well as for elucidating mechanisms underlying aberrant PLS3 expression [149].

## 5. Discussion and Conclusions

Biomarker research in SMA is of great importance for optimizing clinical decision-making and advancing precision medicine. Current investigations have explored a variety of molecular and non-molecular biomarkers (Figure 1); however, their clinical utility varies depending on disease subtype, age, and treatment intervention, and further validation and standardization are needed.

Among molecular biomarkers, *SMN1* genotyping is indeed the primary diagnostic biomarker, and *SMN2* copy number remains the key genetic indicator for predicting disease severity and treatment responsiveness. However, inter-laboratory variability in *SMN2* quantification limits its precision in clinical practice. NFs, particularly pNFH in CSF, have shown significant prognostic value in infantile-onset SMA, correlating strongly with motor improvement during nusinersen treatment. Nevertheless, conflicting data in adolescent and adult populations suggest that the utility of NFs must be re-evaluated in the context of disease stage and through standardized methodologies. MicroRNAs (e.g., miR-133a/b, miR-206) and spatial transcriptomics have illuminated dynamic changes in disease mechanisms and therapeutic responses, opening new avenues for the development of composite biomarker panels. However, their clinical translation requires further validation across multiple centers. While proteins such as amyloid-β and tau may be involved in neuroprotective processes, current evidence is inconsistent, and their specificity as SMA biomarkers remains to be firmly established. Additionally, serum creatinine and creatine kinase levels provide insights into muscle metabolism and damage but are susceptible to confounding factors. Integrating them with other indicators, such as the CCR, may enhance their reliability. Urinary creatinine is a potential indicator for disease prognosis in adult SMA patients. Non-molecular biomarkers, including electrophysiological parameters (e.g., CMAP, MUNE) and imaging modalities (e.g., ultrasound, multispectral optoacoustic tomography [MSOT]), enable non-invasive quantification of motor neuron and muscle pathology. These tools show particular promise in early pediatric diagnosis. However, their clinical implementation is limited by dependence on specialized equipment and the lack of integrated analyses with molecular biomarkers. There are current challenges in biomarker development for SMA. First, longitudinal studies are needed to track the dynamic evolution of biomarkers in the context of disease heterogeneity. Second, fluid-based biomarkers, such as CSF NFs, have limited applicability due to the invasiveness of sample collection. Third, standardized detection protocols and cross-platform validation are also lacking. Future directions should prioritize multi-omics integration (e.g., transcriptomics and proteomics) to identify highly specific biomarkers, develop non-invasive diagnostic technologies (e.g., circulating exosomal miRNAs in blood, and conduct large-scale multicenter studies to establish causal links between biomarkers and clinical outcomes.

Furthermore, these identified biomarkers hold translational potential as targets for the development of targeted therapies. For example, exosomes derived from adipose-derived stem cells (ASC-EVs) exhibit neuroprotective effects in SMA models, improving motor function and delaying motor neuron degeneration. These therapeutic benefits are associated with modulation of apoptotic marker Cleaved Caspase-3 and neuroinflammatory markers GFAP and Ionized calcium binding adaptor molecule 1 (IBA1) [140]. Whole-transcriptome sequencing has revealed that miR-34a and its regulated competing endogenous RNA (ceRNA) network—including Spag5, lncRNA00138536, and circRNA007386—play key roles in both neural and non-neural pathologies of SMA. Inhibition of miR-34a can reverse cell cycle arrest, offering new insights into the mechanisms of non-neural tissue damage in SMA and supporting the development of miR-34a–targeted therapeutic strategies [141]. miR-206 provides multidimensional insights for therapeutic development in SMA. Exogenous supplementation of miR-206 can regulate calcium homeostasis by targeting sodium-calcium exchanger 2 (NCX2), significantly reducing motor neuron loss (by 25% in the brainstem and 15% in the spinal cord), improving motor function, and extending survival by approximately 15% in SMA mouse models. It shows potential as an adjunct to SMN-dependent therapies. The mechanism of action is well defined and combining it with existing treatments may yield synergistic effects. While currently limited by the invasiveness of intraventricular injection, delivery can be optimized using non-invasive systems such as viral vectors (e.g., Adeno-Associated Virus [AAV]), nanoparticles, or intranasal administration. Dynamic expression of miR-206 and its target NCX2 can serve as biomarkers for therapeutic response monitoring. Though still in the preclinical phase, the endogenous nature of miR-206 suggests a favorable safety profile and potential applicability across other neurodegenerative diseases such as ALS. Developing multi-target combination therapies that include SMN activators is a critical future direction [150]. Elevated levels of pro-inflammatory cytokines (e.g., IL-6, TNF-α, MCP-1) and neurotrophic factors (e.g., VEGF, PDGF-BB) in SMA represent potential therapeutic targets. Current treatment with nusinersen shows limited modulation of these factors, indicating the need for combination therapies incorporating anti-inflammatory drugs and CNS-targeted delivery systems [143].

The *SMN2* copy number and SMN protein levels are well-established biomarkers used directly for the diagnosis, prognosis, and therapeutic response prediction in SMN-dependent treatments [144,151]. Although NF proteins and inflammatory cytokines serve as auxiliary indicators for disease monitoring, they have yet to be validated as drug development targets. Current therapeutic strategies remain largely centered on modulating the SMN pathway. However, emerging approaches include combination therapies—such as the co-administration of SMN modulators with anti-inflammatory agents—and the refinement of prenatal gene delivery platforms (e.g., integrase-defective lentiviral vectors [IDLVs), AAV9). Prenatal diagnosis coupled with early intervention, including in utero gene therapy, represents a promising but complex avenue, requiring careful navigation of ethical, safety, and technical challenges [151]. Multi-omics analyses have further identified disruptions in lipid (CE/Chol, Lyso-PL/PL) and amino acid (tryptophan, methionine) metabolism within the cerebrospinal fluid of SMA patients, suggesting novel biomarker candidates for diagnosis, disease progression tracking, and therapeutic monitoring, particularly in the context of nusinersen response [152]. While current interventions do not directly address these metabolic pathways, abnormalities in high-density lipoprotein (HDL) function and sphingolipid metabolism highlight potential targets for future combinatorial therapies.

Despite the predominance of SMN-targeted treatments, no novel molecular targets have been clinically validated to date. Early intervention—especially within the first six weeks of life in presymptomatic individuals—remains pivotal for favorable outcomes and should be guided by an integrated assessment of *SMN2* copy number and clinical presentation [146]. A composite panel including *SMN2* copy number, motor function scores, lipid metabolism markers, and neurofilament proteins offers a more robust framework for diagnosis, disease monitoring, and therapeutic efficacy assessment. Looking ahead, therapeutic paradigms should expand beyond the SMN axis to incorporate metabolic and neuroinflammatory targets. The development of anti-myostatin agents (e.g., Apitegromab, Taldefgrobep alfa) exemplifies this broader strategy. Ultimately, optimizing outcomes in SMA will require a precision medicine approach that combines early diagnosis, multi-dimensional biomarker profiling, and personalized therapeutic regimens tailored to each patient’s genetic and clinical profile [147,148]. By refining the biomarker landscape, we can enable earlier intervention, personalized treatment strategies, and dynamic therapeutic efficacy monitoring in SMA, ultimately improving patient prognosis.

Previous meta-analyses, including data from three transgenic mouse models of SMA and two human SMA studies, have demonstrated altered serum levels of myostatin and follistatin. These studies revealed decreased skeletal muscle myostatin gene expression and increased follistatin expression in SMA transgenic mice, a finding corroborated in the iliopsoas muscle of five patients with SMA type 1 [153]. Clinically, median serum myostatin levels were significantly lower in SMA patients (98 pg/mL; range 5–157) compared to controls (412 pg/mL; range 299–730) (*p* < 0.001). Lower myostatin concentrations correlated strongly with greater disease severity, as assessed by clinician-rated outcomes (Spearman’s Rho = 0.493–0.812; *p* < 0.05). Longitudinal analysis over 12 months showed a further decline in myostatin levels among SMA cases (*p* = 0.021), reinforcing its association with disease progression. In contrast, follistatin levels remained unchanged between groups and over time. However, the follistatin-to-myostatin ratio was significantly elevated in SMA subjects and exhibited an inverse correlation with motor function severity [153]. These findings position serum myostatin as a promising biomarker for assessing SMA severity and progression in type 1 SMA, with potential applications in clinical monitoring and therapeutic evaluation. Another systematic review and meta-analysis (PROSPERO: CRD42021235605) synthesized evidence from 42 studies (606 patients, 19 cohorts) to evaluate potential biomarkers. Pooled analysis confirmed significant differences between SMA patients and controls in upper limb CMAP amplitudes and MUNE values [mean difference −3.63 (−6.2, −1.06), −119.74 (−153.93, −85.56), respectively]. However, contradictory data for other functional scales (e.g., Hammersmith FMSE, RULM, 6-Minute Walk Test [6MWT]) precluded meta-analysis. Given the limited natural history data in treated populations, this study highlights key biomarkers warranting further longitudinal validation to guide clinical assessment and therapeutic monitoring [111].

While this scoping review offers a comprehensive overview of the existing literature on biomarkers SMA, it has notable limitations. Unlike systematic reviews and meta-analyses, which statistically pool data from homogenous studies to quantify effects and assess potential biases, a scoping review does not evaluate the quality of included studies or synthesize quantitative results [154]. The current scoping review organizes evidence thematically, limiting the capacity to establish concrete conclusions regarding the effectiveness of interventions or causal links. Furthermore, the inclusion of varied study designs and broadly framed research queries can result in findings that are less precise than those produced by meta-analyses, which generally concentrate on narrowly defined and comparable outcomes [154]. Systematic reviews, though methodologically robust, are inherently constrained by temporal lags: they capture evidence only up to the search date, and publication delays—often exceeding two years in clinical research—risk obsolescence [154,155]. While updates mitigate this, they require substantial resources without eliminating the latency between search execution and dissemination [154,155]. Nevertheless, scoping reviews serve a critical role in nascent or complex fields like SMA biomarkers, where they delineate conceptual frameworks, identify knowledge gaps, and lay groundwork for future quantitative synthesis. Their utility lies not in replacing systematic reviews, but in catalyzing hypothesis-driven research when high-level evidence remains emergent.

Biomarker research in SMA has advanced significantly, offering critical tools for diagnosis, prognosis, and therapeutic monitoring. While *SMN1* genotyping, *SMN2* copy number and neurofilament proteins remain foundational, emerging biomarkers—such as miRNAs (miR-206, miR-34a), metabolic profiles, and imaging modalities (MSOT, ultrasound)—provide deeper mechanistic insights and potential therapeutic targets. However, challenges persist, including standardization, longitudinal validation, and the need for non-invasive detection methods. Future directions involve multi-omics integration (transcriptomics, proteomics, lipidomics) to refine biomarker panels, non-invasive technologies (circulating exosomal miRNAs, advanced imaging) for broader clinical adoption, combinatorial therapies targeting SMN-independent pathways (e.g., neuroinflammation, calcium homeostasis, lipid metabolism), and early intervention strategies, including prenatal diagnosis and in utero therapy, guided by precision biomarker profiling. By addressing these gaps, SMA management can evolve toward truly personalized medicine, optimizing treatment efficacy and improving long-term outcomes.

## 6. Materials and Methods

A comprehensive literature search was systematically conducted using the PubMed and Web of Science databases to identify relevant publications up to May 2025 (Figure 2). The search strategy incorporated a combination of keywords, including “spinal muscular atrophy” paired with “biomarker,” as well as advanced technologies such as “single-cell omics,” “digital droplet PCR,” “spatial transcriptomics,” “proteomics,” “epigenetics,” “artificial intelligence,” “machine learning,” “nanopore and long-read sequencing,” and “extracellular vesicles.” Titles and abstracts were screened to assess eligibility. Studies were included if they investigated biomarkers associated with SMA’s diagnosis, prognosis, or treatment response. Articles not primarily focused on biomarkers were excluded. No restrictions were applied regarding study design or methodology.

## Figures and Tables

**Figure 1 ijms-26-06887-f001:**
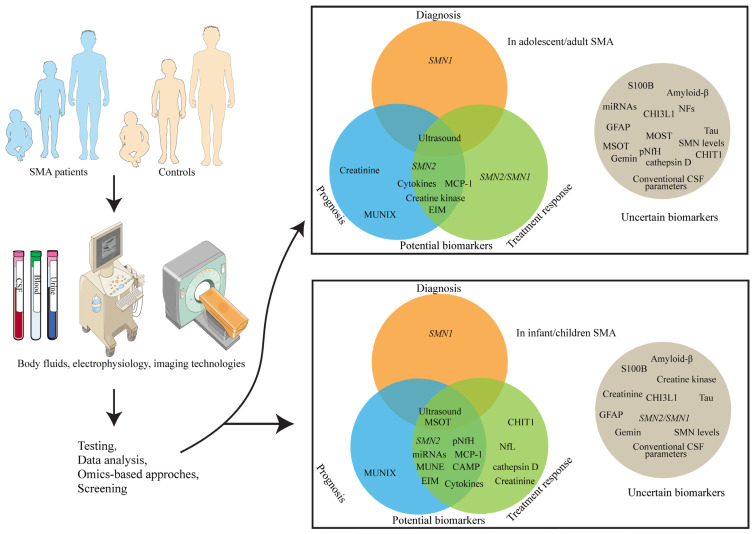
Biological samples, including CSF, blood, and urine, as well as imaging and electrophysiological data, were collected from patients with SMA and matched controls. Omics-based technologies were employed to analyze these samples, enabling high-resolution profiling of molecular and cellular signatures. The findings reaffirmed the *SMN1* gene as the primary diagnostic biomarker for SMA. Additionally, muscle ultrasound, and MSOT emerged as promising biomarkers for diagnosis, prognosis, and therapeutic monitoring. Other potential prognostic and treatment-response biomarkers included *SMN2* copy number, pNfH, miRNAs, MCP-1, creatine kinase, cytokines, creatinine, CMAP, MUNE, and EIM. CHIT1, NfL, cathepsin D, and the SMN2/SMN1 copy number ratio showed promise as a treatment-response biomarker. MUNIX is a potential biomarker for prognosis. However, biomarker profiles differ significantly between pediatric and adolescent/adult SMA patients. SMN: Survival of motor neuron; MOST: Multispectral optoacoustic tomography; pNfH: Phosphorylated neurofilament heavy chain; miRNAs: microRNAs; MCP-1: Monocyte chemoattractant protein-1; EIM: Electrical impedance myography; CHIT1: Chitotriosidase 1; S100B: S100 calcium-binding protein B; CHI3L1: Chitinase-3-like protein 1; GFAP: Glial fibrillary acidic protein; CSF: Cerebrospinal fluid. CMAP: Compound muscle action potential; MUNE: Motor units number estimation; MUNIX: Motor unit number index. This figure was created with elements sourced from Servier Medical Art. Medical Art by Servier is licensed under CC BY 4.0 (https://creativecommons.org/licenses/by/4.0/, accessed on 26 May 2025).

**Figure 2 ijms-26-06887-f002:**
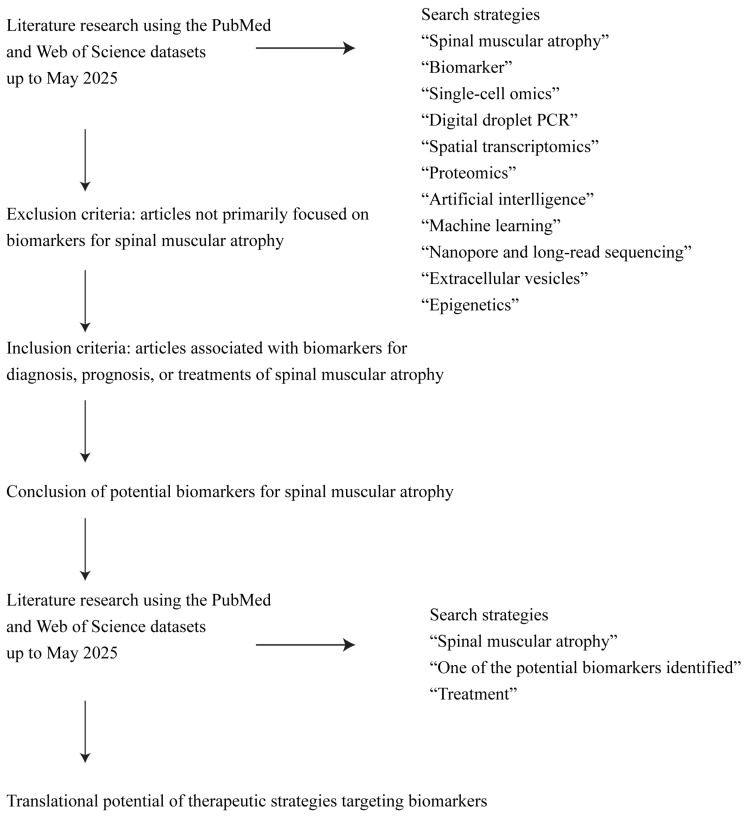
Schematic overview of the research strategy.

**Table 1 ijms-26-06887-t001:** Established and emerging biomarkers for spinal muscular atrophy and their clinical applications.

Biomarkers	Disease Models	Functions	Application of Biomarkers	Reference
*SMN1* gene	SMA patients	Encodes full-length, functional SMN protein; loss or mutation causes SMA	Deletion or mutation of the *SMN1* gene constitutes a diagnostic biomarker, particularly when interpreted alongside clinical features.	[1,10,19]
*SMN2* gene	SMA patients	Produces ~10% full-length SMN protein and ~90% truncated form.	*SMN2* copy number is considered a neuroprotective modifier, inversely correlated with disease severity, and a predictor of treatment response in SMA	[20,21]
*SMN2/SMN1* expression ratio	SMA patients	The *SMN1* gene produces full-length, functional SMN protein, whereas the *SMN2* gene generates predominantly truncated protein, with only ~10% being full-length.	Patients receiving nusinersen who showed clinical improvement exhibited an increased *SMN2*/*SMN1* expression ratio, whereas those with stable disease had a decreased ratio.	[22]
SMN protein or mRNA	SMA mice; SMA patients	Normal SMN protein	The advantage lies in its high specificity, which can directly reflect the functional status of the *SMN1* gene. However, the drawback is that its levels in peripheral blood may be influenced by multiple factors, and it may not fully reflect the changes in the central nervous system.	[20,23,24,25,26,27,28,29,30,31,32,33,34,35,36,37,38,39,40,41]
Neurofilaments (NFs): neurofilament light chain (NfL), neurofilament medium chain (NfM), neurofilament heavy chain (NfH)	SMA mice; SMA patients	Markers of neuronal injury, whose transcript and protein level changes can reflect disease progression and the degree of neuronal injury.	In SMA infants, it can serve as a reliable biomarker to assess prognosis and treatment response to nusinersen. However, its value as a biomarker in adult-onset SMA or during disease progression stages across all SMA types remains unclear.	[30,41,42,43,44,45,46,47,48,49,50,51,52]
Amyloid-β (Aβ40, Aβ42)	Alzheimer’s disease (AD) mice; SMA mice	Aβ is a key constituent of Alzheimer’s plaques, is neurotoxic due to its abnormal aggregation and deposition in the extracellular matrix.	Aβ protein levels (Aβ40, Aβ42, soluble amyloid precursor protein [sAPP]) correlate with motor scores in young SMA patients post-nusinersen treatment but require further validation as biomarkers for treatment response or progression.	[30,53,54,55,56,57,58]
Tau protein	SMA mice SMA patients	Tau, a microtubule-associated protein, plays a key role in facilitating microtubule polymerization and preserving cytoskeletal integrity.	Tau protein shows potential for monitoring nusinersen response in non-adult patients, though inconsistent in adults, requiring further validation in larger multicenter studies. NfL outperforms Tau as a biomarker for monitoring nusinersen treatment response in SMA.	[30,48,55,58,59,60,61,62,63,64,65]
Cerebrospinal fluid (CSF) parameters	SMA patients	Monitors inflammatory status via CSF cell counts/protein/albumin; detects nusinersen-associated “nusinophages” for treatment response insights.	Monitor nonspecific treatment-related inflammation (e.g., post-nusinersen injection) and explore “nusinophages” as potential biomarkers, requiring further clinical validation.	[30,50,63,66,67,68,69,70,71,72]
Cytokines (e.g., interleukin [IL]-8, monocyte chemoattractant protein-1 [MCP-1])	SMA patients	Neuroinflammation-associated cytokines (e.g., IL-8, MCP-1) correlate with SMA severity baseline; track nusinersen treatment response/motor improvements; monitor SMA with progressive myoclonic epilepsy (SMA-PME) progression via plasma MCP-1 changes; suggest biomarker potential for disease regulation.	Cytokines (e.g., IL-8, MCP-1) assess SMA-PME disease status/treatment efficacy; confirm neuroinflammation pathophysiology via dysregulation; screen subtype-specific cytokine panels (SMA1, SMA-PME, adult SMA) for precision diagnosis/mechanistic research.	[30,73,74,75]
Creatine kinase (CK)	SMA patients	Muscle injury and rhabdomyolysis	The advantage is that it is readily measurable and directly correlated with muscle injury. It is related to disease severity and treatment response in adult SMA. However, its evaluation in pediatric SMA requires further investigation.	[30,76,77,78,79]
Creatinine (Crn)	SMA patients	Serum Crn is the end product of creatine metabolism in skeletal muscle, and its levels can reflect the functional status and energy metabolism of skeletal muscle.	Serum Crn shows promise as a biomarker for disease severity in adult and adolescent SMA patients, while urinary Crn shows promise as a biomarker for monitoring treatment response in SMA Type 1. However, more research is needed to validate serum Crn in pediatric populations, particularly infants and young children.	[30,76,80,81]
Chitotriosidase 1 (CHIT1) and CHI3L1/YKL-40	SMA patients	Reflects the proliferation of microglia and astrocytes.	The advantage lies in its role as a biomarker of neuroinflammation, potentially reflecting the disease’s inflammatory status. CHIT1 appears to be a promising CSF biomarker for tracking treatment response in pediatric SMA, while CHI3L1 shows inconsistent changes across studies and no significant alterations in slowly progressing adult SMA patients treated with nusinersen.	[30,65,70,82,83,84]
Glial fibrillary acidic protein (GFAP)	SMA patients	Glial cell activation and neuroinflammation-	The advantage lies in its role as a biomarker of glial cell activation, potentially reflecting the disease’s inflammatory status. However, its levels may be influenced by multiple factors.	[30,48,51,85,86]
S100 calcium-binding protein B (S100B)	SMA patients	Predominantly localized in astrocytes of the nervous system, it serves as a reliable biomarker reflecting neural injury.	The advantage lies in its role as a biomarker of neural injury, potentially reflecting disease progression. However, its performance may vary across studies.	[30,59,62,87,88]
MicroRNAs (miRNAs)	SMA cell-based experiment; SMA patients	Regulate gene expression and disease progression	miRNAs show strong potential as biomarkers for tracking SMA progression and treatment response, particularly in pediatric and infant populations. However, more research is needed to establish their utility in adult SMA patients due to limited supporting data.	[30,89,90,91,92,93,94,95,96,97,98,99,100,101]
Gemin proteins	SMA zebrafish, *Drosophila*, and mice, human induced pluripotent stem cell-derived motor neurons	Serve as crucial chaperones for the SMN protein to regulate key steps in small nuclear ribonucleoprotein (snRNP) assembly precisely	The stability of the Gemin complex, particularly spinal Gemin2 levels, and its efficiency in snRNP assembly are linked to splicing dysfunction in SMA patients. However, Gemin proteins have limitations as SMA biomarkers due to low specificity, detection challenges, and insufficient clinical correlation.	[102,103,104,105,106,107]
Compound muscle action potential (CMAP) and MUNE (motor units number estimation)	SMA patients	Motor unit number/function assessment is critical for SMA diagnosis, disease monitoring, treatment evaluation, and clinical decision-making.	CMAP amplitude measures motor unit function to assess SMA severity and treatment response; MUNE methods (motor unit number index [MUNIX], multipoint) explore SMA applications but face limitations from proximal muscle weakness.	[41,108,109,110,111,112,113,114]
Electrical impedance myography (EIM)	SMA patients	EIM is a bioimpedance-based technology that is sensitive to muscle changes.	Due to its rapid, non-invasive, quantitative, and painless characteristics, this technology is highly suitable for tracking pediatric neuromuscular diseases. Additionally, it has previously been studied in older children with SMA.	[115]
Ultrasound	SMA patients	Quantifies muscle structural changes (e.g., increased echogenicity [EI], reduced muscle thickness [MT], moth-eaten heterogeneity) for disease assessment.	Muscle ultrasound identifies SMA patients (e.g., type 1) via EI deviations, quadriceps fat thickening, and mixed atrophy/hypertrophy; monitors progression in type 1 infants through parameter decline; evaluates treatment response (e.g., nusinersen) non-invasively. It also shows promise as a biomarker for diagnosing, monitoring disease progression, and assessing treatment response in adult SMA. However, larger validation studies are required to establish its clinical utility.	[116]
Multispectral optoacoustic tomography (MSOT)	SMA patients	Enables multispectral tissue imaging and quantitative analysis via photoacoustic ultrasonic signals generated from near-infrared laser absorption variations.	Photoacoustic signals assess SMA severity via disrupted muscle patterns and dispersed/diminished signals in ambulatory patients. MSOT holds promise for assessing pediatric SMA, but its utility in adult patients remains uncertain and requires further validation across age groups and disease stages.	[117]

**Table 2 ijms-26-06887-t002:** Five clinical types of spinal muscular atrophy.

**Clinical Type**	***SMN2* Copy Number**	**Clinical Features**
SMA Type 0	1	⬤ Onset before birth with reduced fetal movement in late pregnancy. ⬤ At birth, infants show severe muscle weakness and hypotonia. Except for eye movement, there is almost no activity of the limbs, trunk, or facial muscles. No sucking reflexes, presence of congenital joint contractures, muscle atrophy, and absence of reflexes. ⬤ Respiratory support is required immediately after birth. ⬤ Congenital heart defects may also be present. ⬤ Most infants die within weeks.
SMA Type 1	1 or 2	⬤ Onset before 6 months of age. ⬤ The most common and severe infantile-onset form. ⬤ Affected infants typically present as “floppy infant syndrome” with severe generalized hypotonia. Due to weakness of the tongue, facial, and chewing muscles, there is difficulty in sucking and swallowing. Tongue atrophy and fasciculations are common. Motor weakness primarily manifests in the lower extremities, with greater proximal musculature involvement than distal muscle groups. Infants cannot control head movements, roll over, or sit independently. ⬤ Respiratory compromise arises early due to severe intercostal muscle weakness, resulting in paradoxical abdominal breathing and the development of a bell-shaped thorax. ⬤ As the disease progresses, voluntary motor function continues to decline. The majority of patients develop respiratory failure secondary to recurrent pulmonary infections, with approximately 80% dying before one year of age. A small proportion may survive beyond two years with intensive supportive care.
SMA Type 2	3	⬤ Onset between 6 and 18 months. Normal development during the first 6 months, followed by stagnation of motor development. ⬤ The hallmark feature is progressive, symmetric muscle weakness and hypotonia, with greater involvement of proximal than distal muscles and lower limbs more severely affected than upper limbs. ⬤ Children can usually sit independently but are unable to stand or walk unaided. Clinical examination reveals limb weakness, decreased or absent deep tendon reflexes, and characteristic signs such as tongue atrophy and fasciculations. Postural hand tremors are observed in approximately 50% of patients. ⬤ Over time, musculoskeletal complications become more prominent. Progressive scoliosis, large joint contractures, and restrictive pulmonary insufficiency are common, often requiring orthopedic and respiratory interventions. Despite the severity of motor impairment, cognitive and language development remain normal. ⬤ With comprehensive multidisciplinary care, most individuals with SMA Type 2 survive into adolescence or early adulthood, though respiratory complications are a leading cause of morbidity and mortality.
SMA Type 3	3 or 4	⬤ Onset between 18 months and 10 years of age, typically after a period of normal early motor development. ⬤ Symptoms begin in childhood or adolescence, with initial symptoms often including difficulty running, climbing stairs, or rising from the floor. Patients can walk independently but experience gradually progressive proximal muscle weakness, more in the lower limbs than the upper limbs. In the early stages, weakness may be segmental. Spinal deformities, joint abnormalities, and respiratory insufficiency may occur in later stages. Despite the progressive motor impairment, intellectual and cognitive development remain intact. ⬤ Life expectancy is near normal, with most individuals surviving into mid-to-late adulthood, especially with appropriate supportive care and access to disease-modifying therapies.
SMA Type 4	4 or more	⬤ Adult onset typically occurs between 30 and 60 years of age. ⬤ Characterized by slowly progressive, predominantly proximal muscle weakness, with greater involvement of the shoulder and pelvic girdle musculature. ⬤ Symptoms may include difficulty with activities requiring proximal strength, such as climbing stairs, rising from a seated position, or lifting objects overhead. Bulbar and respiratory muscles are usually spared, and functional mobility is often preserved for many years. ⬤ The disease course is indolent, and while it may cause disability over time, it does not significantly impact life expectancy. Cognitive function remains normal. Due to its subtle and late onset, Type 4 may be underdiagnosed or misattributed to other adult neuromuscular disorders.

## Data Availability

All datasets are available in the main text and will be available from the corresponding author upon request.

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
