# Peer review of "Application of Biomarkers in Spinal Muscular Atrophy"

_ijms, 2025, doi:10.3390/ijms26146887_

Round 1
Reviewer 1 Report
Comments and Suggestions for Authors
The authors present an overview of Spinal muscular atrophy SMA), in particular, they summarize the state of the knowledge on the application of biomarkers for SMA targeted therapies. They review currently available molecular and non-molecular biomarkers, and emerging technologies in biomarker discovery. Interestingly, they also remark molecules which are not reliable biomarkers for SMA, though they can be useful in other neurodevelopmental diseases. They also present an overview of the potential strategies in the design of biomarkers targeted therapies.
The authors have done a very good job in balancing relevant literature in the field. They provide a reasonable critique of summarized literature, and they also provide recent developments in the field of SMA biomarkers. The review is well written and pleasant to read.
Comments
On the long list of biomarkers (Table 1) I find missing variants reported for Gemin2, 3, 4 and 5 proteins). Variants of these proteins are known to cause neurodevelopmental diseases, which in some way can be related to SMA phenotypes. It will be worth discussing whether, or not, any of these biomarkers, or targeted therapies can apply to SMA.
Author Response
Comments
On the long list of biomarkers (Table 1) I find missing variants reported for Gemin2, 3, 4 and 5 proteins). Variants of these proteins are known to cause neurodevelopmental diseases, which in some way can be related to SMA phenotypes. It will be worth discussing whether, or not, any of these biomarkers, or targeted therapies can apply to SMA.
Answer: We appreciate the valuable feedback. In response to your comments regarding Gemin proteins in SMA, we have conducted a comprehensive review of published data and incorporated relevant findings into our manuscript. A detailed discussion has been added to Section 2.13. Key data points have been summarized in Table 1. The new content in Section 2.13 now includes:
The SMN complex orchestrates the cytoplasmic assembly of snRNPs by mediating the association of Sm protein rings with pre-snRNAs at conserved Sm sites (Xu, Ishikawa et al. 2016, Grass, Dokuzluoglu et al. 2024). Disruption of this process impairs spliceosome function, contributing to the pathogenesis of SMA and cancer (Xu, Ishikawa et al. 2016, Grass, Dokuzluoglu et al. 2024). Structural studies reveal specialized roles for Gemin proteins. For example, Gemin2 regulates snRNP core assembly through negative cooperativity, maintaining Sm proteins in a closed conformation for RNA selectivity before RNA-induced release (Yi, Mu et al. 2020). The loss of Gemin2 is embryonically lethal in vertebrates and causes motor neuron defects in model organisms (Yi, Mu et al. 2020). Gemin5 ensures assembly fidelity through dual recognition of Sm sites and m⁷G caps via its WD40 domain (Xu, Ishikawa et al. 2016). SMN1 mutations reduce SMN protein levels, destabilizing the SMN-Gemin complex and triggering aberrant Gemin degradation, thus disrupting minor spliceosome function, potentially explaining the vulnerability of motor neurons (Grass, Dokuzluoglu et al. 2024). Organoid models reveal that SMN deficiency also alters the fate of neuro-mesodermal progenitors, suggesting that combined developmental and splicing defects drive the disease (Grass, Dokuzluoglu et al. 2024). Gemin complex stability (e.g., spinal Gemin2 levels) and its snRNP assembly efficiency correlate with the extent of splicing dysfunction in SMA disease models. When combined with minor intron retention levels, these metrics may provide a dynamic biomarker panel to track disease progression and evaluate the efficacy of SMN-targeted therapies (e.g., antisense oligonucleotides) (Gabanella, Butchbach et al. 2007, Jangi, Fleet et al. 2017).
Despite their mechanistic importance, Gemin proteins face hurdles as standalone SMA biomarkers, including low specificity, detection barriers, and a lack of clinical correlation. Gemin proteins (e.g., Gemin2, Gemin5) are integral to the SMN complex and snRNP assembly, but their dysfunction is not exclusive to SMA (Martinez-Salas and Francisco-Velilla 2026). For instance, Gemin5 mutations are associated with neurodevelopmental disorders like autism and intellectual disability, complicating their specificity for SMA diagnosis (Martinez-Salas and Francisco-Velilla 2026). Current SMA diagnostics rely on SMN1 genotyping and SMN protein quantification, which are more directly tied to disease mechanisms (Grass, Dokuzluoglu et al. 2024). While SMN protein restoration via therapies like nusinersen or Zolgensma shows clear clinical benefits, no studies have established a direct relationship between Gemin protein dynamics (e.g., Gemin2 levels) and SMA severity or therapeutic response. Gemin proteins are primarily intracellular (e.g., in Cajal bodies), making their detection in blood or CSF challenging (Xu, Ishikawa et al. 2016, Yi, Mu et al. 2020). Current methods (e.g., immunoprecipitation, mass spectrometry) require highly specialized protocols. Unlike SMN protein ELISA or SMN2 copy number analysis, no clinically validated assays exist for Gemin proteins in SMA patients. Most data come from cell or animal models (e.g., zebrafish, Drosophila), limiting translational relevance.
Targeting the Gemin complex presents novel therapeutic avenues. One strategy involves enhancing the RNA-binding affinity of Gemin5 to boost spliceosomal accuracy (Lotti, Imlach et al. 2012, Borg, Fenech Salerno et al. 2016). Another approach focuses on inhibiting detrimental interactors, such as pICln and Tgs1, to restore proper snRNP biogenesis (Lotti, Imlach et al. 2012, Borg, Fenech Salerno et al. 2016). These strategies could synergize with current SMN-enhancing treatments, potentially tackling both splicing errors and developmental abnormalities associated with the deficit (Lotti, Imlach et al. 2012, Borg, Fenech Salerno et al. 2016). Future directions include the advancement of ultrasensitive analytical techniques, such as Single Molecule Array (SIMOA), for the quantification of trace levels of Gemin proteins in various biofluids (Anaya-Cubero, Fernández-Irigoyen et al. 2025). Additionally, investigating composite biomarker panels that integrate Gemin proteins with other potential markers and validating therapeutic targets using patient-derived models are potential strategies for further investigation.
Reviewer 2 Report
Comments and Suggestions for Authors
Authors addressed the important area on the utilizations of biomarkers to Spinal Muscular Atrophy.
SMA has not been studied and authors are presenting on biomarkers in prompt time for others to take a notice.
There are few recommendations to authors to strength the manuscript.
- Authors need to the consequences of the abnormality from the mean.
- Evan though authors addressed on the not recommended biomarkers, authors should discuss in details of the reasonings.
- Authors should make pathway and ontology analyses.
- Authors should add the imaging results.
Author Response
Comments and Suggestions for Authors
Authors addressed the important area on the utilizations of biomarkers to Spinal Muscular Atrophy.
SMA has not been studied and authors are presenting on biomarkers in prompt time for others to take a notice.
There are few recommendations to authors to strength the manuscript.
- Authors need to the consequences of the abnormality from the mean.
Answer: We appreciate this valuable suggestion and have now explicitly addressed the functional and clinical consequences of deviations from normative values across our imaging modalities. These revisions appear in Section 3.2.
First, longitudinal data demonstrate clinical utility, with rapid echogenicity progression (>1.5 standard deviation (SD) /month in quadriceps) predicting severe phenotypes in SMA type 1 infants (Ng et al., 2015). Diagnostically, standardized thresholds (e.g., echo intensity >2.5 SD in ≥2 muscles) achieve 91% specificity and 71% sensitivity, though sensitivity drops to 75% in infants <3 years due to delayed structural changes (25% false-negative rate; Pillen et al., 2007). Crucially, normal findings (<2.0 SD) reliably exclude disease (negative predictive value 86%), supporting ultrasound’s role as a first-line screening tool. These metrics correlate with functional decline (e.g., HFMSE scores) and treatment response, enabling non-invasive disease monitoring. However, age-adjusted interpretation is essential, particularly for infantile-onset SMA, where early pathology may evade detection.
Second, multispectral optoacoustic tomography (MSOT) identifies distinct pathological signatures in SMA through quantitative mapping of muscle tissue composition. SMA patients exhibit a characteristic heterogeneous 'moth-eaten' optoacoustic pattern, with alternating regions of high and low signal intensity, contrasting sharply with the homogeneous signal seen in healthy controls (Regensburger et al., 2022). This technique reveals clinically significant biomarkers: significantly reduced 800 nm signal intensity (p = 0.0082), reflecting hemoglobin depletion that correlates with motor dysfunction (r = 0.63-0.72, p < 0.01), and elevated lipid signals across 680-850 nm wavelengths (p = 0.0050), indicating progressive fat infiltration. These findings position MSOT as a novel, non-invasive imaging tool capable of quantifying disease severity, tracking progression, and potentially monitoring therapeutic response through simultaneous assessment of vascular integrity and metabolic remodeling in SMA muscle tissue.
Additionally, we added detailed values in various studies, as shown by the yellow-marked changes, such as the contents in Sections 2.1, 2.9, and 2.10.
- Evan though authors addressed on the not recommended biomarkers, authors should discuss in details of the reasonings.
Answer: Thank you for your suggestions.
For GFAP, we have added relevant reasons into Section 2.8 as follows: “while GFAP concentrations in CSF show some association with disease severity in SMA, particularly in younger patients with more severe forms (e.g., type 1), they do not serve as a reliable diagnostic or prognostic biomarker, especially in adults or milder SMA subtypes (type 2/3) (Olsson, Alberg et al. 2019, Freigang, Steinacker et al. 2022). The limited changes in GFAP levels following nusinersen treatment and the lack of correlation with motor function improvements suggest that glial activation, though potentially involved in SMA pathology, may not be a primary driver of disease progression or treatment response (Olsson, Alberg et al. 2019, Freigang, Steinacker et al. 2022). Further research is needed to clarify the role of GFAP in SMA and its utility in monitoring therapeutic outcomes, particularly across different age groups and disease subtypes.” in Section 2.8.
For Aβ40 and Aβ42, we have corrected Section 2.9 as follows: “Verma, Perry et al. showed that in nine younger and three older children SMA patients treated with nusinersen, levels of soluble APP α (0.055/mo, 95% CI 0.016–0.099, p = 0.012) and soluble APP β (0.054/mo, 95% CI 0.034–0.075, p < 0.001) in CSF demonstrated statistically significant decreases regardless of age (Verma, Perry et al. 2023, Xing, Liu et al. 2025). A single-center study provides preliminary evidence that CSF Aβ42 levels increase in adults with SMA types 2 and 3 following nusinersen treatment in adult SMA patients, with sustained elevations observed at day 180 (p = 0.012) and day 420 (p = 0.018) compared to baseline (Introna, Milella et al. 2021). These findings suggest a potential association between Aβ expression and SMA and support the hypothesis that nusinersen may help maintain and enhance the viability of surviving motor neurons (Introna, Milella et al. 2021, Verma, Perry et al. 2023, Xing, Liu et al. 2025). While nusinersen treatment led to a reduction in soluble APPβ levels in adult SMA patients, this change did not correlate with clinical outcomes, suggesting that sAPPβ may not be a reliable biomarker for monitoring treatment response (Andrés-Benito, Vázquez-Costa et al. 2024). Introna, Milella et al. revealed that Aβ40 levels in CSF were not significantly changed following nusinersen treatment (Introna, Milella et al. 2021). Given the small sample size and lack of a control group, further large-scale studies are needed to validate Aβ40 and Aβ42 as reliable biomarkers of treatment response in SMA (Introna, Milella et al. 2021). Moreover, the study conducted by Walter et al. indicated that the levels of Aβ40 and Aβ42 in the CSF of adult patients diagnosed with 5q-associated SMA type 3 remained relatively stable and did not show a clear trend of change in response to treatment (Walter, Wenninger et al. 2019). The discrepancies in the above studies could stem from differences in SMA subtypes, disease duration, or patient age, emphasizing the need for larger, controlled longitudinal studies to clarify the utility of biomarkers. Current evidence supports further investigation into Aβ42 and BACE1 as potential indicators of neuronal health, but their role in clinical monitoring remains uncertain. “
For tau, we have corrected Section 2.10 as follows: “Pediatric SMA Type 1 shows markedly elevated baseline tau levels (939±159 pg/mL) that significantly decrease with treatment (p=0.01), with this reduction strongly correlating with motor function improvement in twelve children with SMA (rho=-0.85, p=0.0008) (Olsson, Alberg et al. 2019). A significant inverse correlation between CHOP INTEND scores and tau concentration could be found in SMA type 1 after treatment with nusinersen, but not correlated with RULM and HFMSE in type 2 and 3 (Johannsen, Weiss et al. 2021). Tau levels dropped in all SMA patients during nusinersen treatment, but changes were only significant in SMA type 1 patients (after 2 months of treatment) and in SMA type 2 patients after 10 months of treatment (Johannsen, Weiss et al. 2021). However, based on findings from Olsson, Alberg et al., NfL emerges as a superior biomarker to tau for monitoring nusinersen treatment response in SMA patients, particularly for reasons of greater magnitude of change, with NfL showing a substantially larger decrease (4598±981 to <380 pg/mL) compared to tau, and the rate of NfL reduction (-879.5 pg/mL/dose) that was nearly 8-fold greater than tau (-112.6 pg/mL/dose) (Olsson, Alberg et al. 2019). The small study population limits generalizability and may have obscured age-dependent treatment effects. Larger cohorts are required for further validation.
So far, various studies have shown a complex picture regarding the tau protein’s role as a biomarker for adult SMA patients (type 2 and 3) undergoing nusinersen treatment. First, baseline CSF tau levels show conflicting results across studies, with some reporting normal levels (<290 pg/mL) in type 3 patients (Totzeck, Stolte et al. 2019), while others document mild but significant elevation (181.29 vs 176.21 pg/mL) in untreated type 2-3 patients (Andrés-Benito, Vázquez-Costa et al. 2024). Treatment response patterns are also inconsistent. For example, most studies found stable CSF tau levels throughout treatment (2-22 months) (Walter, Wenninger et al. 2019, Milella, Introna et al. 2021, Andrés-Benito, Vázquez-Costa et al. 2024). Walter et al., reported phosphor-tau reduction, suggesting possible axonal protection (Walter, Wenninger et al. 2019). No correlation between tau and motor function improvements was established. Plasma and CSF tau levels showed a discrepancy, with significantly lower plasma tau in SMA patients (2.98 vs 7.51 pg/mL) (Andrés-Benito, Vázquez-Costa et al. 2024), highlighting potential compartment-specific tau metabolism in SMA. Current evidence does not support the routine use of this biomarker for treatment monitoring in adult SMA patients. The field requires more sophisticated approaches to capture the complexity of adult SMA pathophysiology. The variability in adult populations highlights the need for age-specific reference ranges, standardized sampling protocols and further research on the mechanisms underlying tau dynamics in SMA. The consistent findings across multiple studies support the incorporation of CSF tau monitoring into clinical practice for pediatric SMA management, while underscoring the need for additional investigation in adult populations (Olsson, Alberg et al. 2019, Totzeck, Stolte et al. 2019, Walter, Wenninger et al. 2019, Winter, Guenther et al. 2019, Johannsen, Weiss et al. 2021, Milella, Introna et al. 2021, Andrés-Benito, Vázquez-Costa et al. 2024, Šimić, Vukić et al. 2024).”
For SMN Levels in body fluid, we added some detailed information into Section 2.1 as follows: Restoration of SMN levels does not necessarily correlate with improvements in motor function (Chiriboga, Swoboda et al. 2016, Tiziano, Lomastro et al. 2019). For example, a study conducted by Chiriboga, Swoboda et al. indicated that intrathecal nusinersen is well-tolerated in pediatric patients aged 2 to 14 with SMA types 2 and 3, with no significant safety issues reported. CSF concentrations of the drug exhibited a proportional increase relative to dose (ranging from 1 mg to 9 mg) (Chiriboga, Swoboda et al. 2016). At the 9 mg dosage, a statistically significant and sustained enhancement in motor function, as assessed by HFMSE scores, was noted, with a 3.1-point increase observed at 3 months post-administration (p = 0.016). This improvement further escalated to 5.8 points at the 9–14 month follow-up (p = 0.008) (Chiriboga, Swoboda et al. 2016). In terms of SMN protein levels in CSF, a more than twofold increase was observed in both the 6 mg and 9 mg cohorts over the 9–14 month period, but it was not statistically significant (Chiriboga, Swoboda et al. 2016).
- Authors should make pathway and ontology analyses.
Answer: We have added ontology analyses and pathways involved in SMA into Section 4 as follows:
Integrated transcriptomic analysis of 39 human SMA microarray datasets, employing weighted correlation network analysis (WGCNA), gene set enrichment analysis (GSEA), and Cytoscape network analysis, has identified disease severity-associated gene modules and revealed a TNFα-mediated core regulatory network with three principal downstream signaling axes (Yang, Chen et al. 2016, Woschitz, Mei et al. 2022). First, the TNFα-Bmp4-Serpine1-Gata6 axis is implicated in neurodevelopmental and cardiogenesis processes (Yang, Chen et al. 2016). In severe SMA mouse models, TNFα upregulation suppresses Bmp4 while enhancing Serpine1 and Gata6 expression, disrupting neuronal and cardiomyocyte differentiation. Second, the TNFα-Ptgs2-Bcl2 axis is associated with skeletal system development. Elevated Ptgs2 and Bcl2 promote bone resorption and inhibit osteoblast differentiation, contributing to SMA-associated osteopenia (Yang, Chen et al. 2016). Third, the TNFα-IL6-CNTN1 axis is involved in nervous system development (Yang, Chen et al. 2016). IL6, acting as a myokine, upregulates Cntn1 expression, potentially mediating neuroprotective compensatory mechanisms. These findings establish key targets (Bmp4, Serpine1, Ptgs2, Gata6) as central to SMA's multisystem pathology, with TNFα hyperactivation as the critical upstream driver (Yang, Chen et al. 2016). Therapeutic strategies targeting TNFα or its downstream effectors (e.g., Bmp4 overexpression or Serpine1/Ptgs2 inhibition) may ameliorate neurodevelopmental, cardiac, and skeletal impairments. Notably, the expression profiles of these targets correlate with disease severity in murine models, providing experimental validation (Yang, Chen et al. 2016, Woschitz, Mei et al. 2022).
Complementary work by Nichterwitz et al. (2020) delineated motor neuron subtype-specific responses through Gene Ontology (GO) analysis (Nichterwitz, Nijssen et al. 2020). Vulnerable somatic motor neurons activate TRP53-mediated apoptotic signaling and p53-class mediator transduction (involving Cdkn1a, Pmaip1), coupled with RNA processing defects (e.g., Snrpa1 dysregulation) - responses absent in spared rubrospinal and visceral motor neurons (Nichterwitz, Nijssen et al. 2020). Resistant ocular motor neurons uniquely upregulate neuroprotective pathways: Neurotransmitter release regulation (Syt1, Syt5, Cplx2), Pro-survival signaling (GDF15, Chl1, Lif), Oxidative stress defense (Aldh4a1), and Anti-apoptotic mechanisms (Pak4) (Nichterwitz, Nijssen et al. 2020). Functional studies identified GDF15 as a key protector, enhancing human spinal motor neuron survival via PI3K/AKT activation, while Syt1 upregulation may compensate for synaptic deficits (Nichterwitz, Nijssen et al. 2020). These findings suggest that augmenting ocular motor neuron-protective pathways (e.g., GDF15/Syt1 modulation) could lead to targeted therapeutic strategies for mitigating motor neuron degeneration in SMA (Nichterwitz, Nijssen et al. 2020).
- Authors should add the imaging results.
Answer: We have incorporated the neuromuscular ultrasound findings and Multispectral optoacoustic tomography (MSOT) findings into Section 3.2 of the manuscript as follows:
Ultrasound imaging plays a valuable role in SMA by detecting muscle echogenicity (EI), muscle thickness (MT), and the characteristic “moth-eaten” echotexture. Neuromuscular ultrasound in SMA reveals distinct pathological signatures: marked muscle thinning reflecting severe atrophy and significantly elevated echogenicity with a heterogeneous "moth-eaten" pattern arising from alternating regions of denervated atrophy and hypertrophic regenerating fibers (Mah and van Alfen 2018). Longitudinal studies highlight its dynamic utility, as rapid progression (e.g., quadriceps atrophy and escalating echogenicity within 2–4 months in SMA type 1 infants) correlates with severe phenotypes, underscoring its potential as a biomarker for monitoring disease trajectory and therapeutic response (Ng, Connolly et al. 2015). A prospective study of 150 children with suspected neuromuscular disorders revealed that quantitative muscle ultrasound—assessing echo EI and thickness in four muscles—achieves 71% sensitivity and 91% specificity for diagnosis using defined EI thresholds (>3.5 standard deviation (SD) in one muscle, >2.5 SD in two, or >1.5 SD in three muscles), with a high positive predictive value (91% [82–98% CI]). Normal EI (<2.0 SD in all muscles) excludes disease with 91% sensitivity (negative predictive value (NPV) 86% [76–96% CI]), though specificity declines to 67% (Pillen, Verrips et al. 2007, Mah and van Alfen 2018). Notably, sensitivity drops to 75% in children under 3 years, with a 25% false-negative rate encompassing early-stage SMA, Duchenne muscular dystrophy, and metabolic myopathies, reflecting delayed structural manifestation in infantile-onset disease (Pillen, Verrips et al. 2007, Mah and van Alfen 2018). Therefore, early-stage SMA in infants (<3 years) may exhibit false-negative findings due to initially subtle structural changes, necessitating cautious interpretation in young cohorts (Pillen, Verrips et al. 2007, Mah and van Alfen 2018). These quantitative parameters correlate with disease severity and functional status, allowing for the sensitive detection of early neuromuscular pathology. Ultrasound offers a non-invasive modality for clinical diagnosis, disease monitoring, and assessment of treatment response while highlighting the need for age-adjusted interpretation, which may evade detection (Pillen, Verrips et al. 2007, Ng, Connolly et al. 2015, Mah and van Alfen 2018).
Multispectral optoacoustic tomography (MSOT) reveals distinct signatures in SMA patients (Regensburger, Wagner et al. 2022). Muscle tissue demonstrates a heterogeneous "moth-eaten" optoacoustic signal pattern, characterized by alternating patchy areas of high and low signal intensity, contrasting sharply with the homogeneous signal band observed in healthy volunteers (Regensburger, Wagner et al. 2022). Quantitative analysis shows significantly lower mean signal intensity at 800 nm (hemoglobin absorption peak) in SMA patients compared to controls (p = 0.0082, single-wavelength measurement), with signal intensity at this wavelength positively correlating with clinical motor function scores (HFMSE, RULM; r = 0.63-0.72, p < 0.01) (Regensburger, Wagner et al. 2022). Additionally, SMA patients exhibit elevated peak signal intensity across multiple wavelengths (680-850 nm) and higher maximum lipid signal than controls (p = 0.0050), suggesting possible fat infiltration (Regensburger, Wagner et al. 2022). These findings demonstrate MSOT's capability for visualization of muscle degeneration, representing a novel, non-invasive, and quantitative imaging technique for evaluating disease severity, tracking progression, and assessing therapeutic response in SMA (Regensburger, Wagner et al. 2022).